

# CoSWAT-WQ v1.0: a high-resolution community global SWAT+ water quality model

Albert Nkwasa[1,2], Celray James Chawanda[2,3], Maria Theresa Nakkazi[1], Ann van Griensven[2,4]

[1] Water Security Research Group, Biodiversity and Natural Resources Program, International Institute for Applied
Systems Analysis (IIASA), Schlossplatz 1, A-2361, Laxenburg, Austria

[2] Department of Water and Climate, Vrije Universiteit Brussel (VUB), 1050 Brussel, Belgium

[3] Texas A&M AgriLife Research, Blackland Research & Extension Center, Temple, TX 76502, USA

[4] Water Science & Engineering Department, IHE Delft Institute for Water Education, 2611 AX Delft, The
Netherlands

*Corresponding author: Albert Nkwasa (nkwasa@iiasa.ac.at)

**Abstract**

Quantifying the global extent of anthropogenic impacts on freshwater quality remains challenging due to limited
monitoring data, especially in low and middle-income regions. To address this gap and improve our understanding
of surface water quality, we introduce CoSWAT-WQ, a large-scale water quality model developed to simulate river
water quality constituents of Total Nitrogen (TN) and Total Phosphorus (TP), across global and regional freshwater
systems. CoSWAT-WQ, an adaptation of the Soil and Water Assessment Tool (SWAT), is run at a global scale,
providing high-resolution simulations that capture spatial and daily temporal dynamics of riverine nutrient loads.
Here, we describe the model's inputs, setup structure, processes, and evaluate its performance by comparing model
outputs to in-situ water quality observations and other global nutrient models. CoSWAT-WQ achieves comparable
ranges of river nutrient loads in comparison to other global nutrient models. Additionally, a normalized root mean
square error (nRMSE) < 1 was achieved with in-situ observations at more than 80 % of the gauging stations for
TN and TP concentrations. However, there was a general weak underestimation of observed concentrations and
variability as seen with low Kling–Gupta efficiency (KGE) values for selected stations. Despite its limitations, the
model enables the simulation of river TN and TP constituents at a global scale while keeping local relevance.
CoSWAT-WQ's modular setup allows coupling with sectoral models addressing lake systems, agricultural runoff,
and aquatic biodiversity, thereby broadening its applicability for cross-sectoral assessments. The model outputs
offer valuable data that can inform ecological risk assessments, human health evaluations, and policy decisions on
global freshwater quality management.

**Keywords**: Total Nitrogen, Total Phosphorus, Global scale, SWAT+, Freshwater



## 1. Introduction

Water pollution is a growing global concern, with freshwater and coastal ecosystems increasingly threatened by a wide range of pollutants, leading to deteriorating water quality (UNEP, 2016; du Plessis, 2022). While most pollution emissions originate locally within terrestrial systems, their effects spread regionally and globally through water and air circulation, impacting ecosystems far beyond their source (Bouwman et al., 2006). Addressing water quality degradation is a critical global imperative, yet one of the greatest challenges is understanding the full extent

of the crisis and reducing associated uncertainties. This knowledge gap complicates our ability to predict where the impacts will be most severe and assess the effectiveness of protection and restoration efforts. Without accurate and comprehensive data, we risk compromising the health and livelihoods of the most vulnerable populations (UNEP, 2024). For example, nutrient pollution (Nitrogen - N and phosphorus - P) is one of the major pressures on global aquatic ecosystems altering their condition leading to profound changes in aquatic biodiversity and

biogeochemical processes (Glibert, 2017).

Nitrogen (N) and phosphorus (P) are essential nutrients that support plant and algae growth in aquatic ecosystems, thereby sustaining fish production. However, when present in excessive amounts, nutrient inputs can lead to eutrophication, where the negative impacts often outweigh the benefits, causing a decline in water quality along the freshwater to marine continuum (Wurtsbaugh et al., 2019). In its most severe forms, eutrophication triggers

algal blooms, algal scum, extensive benthic algal growth, oxygen depletion, and the proliferation of submerged and floating macrophytes (Smayda, 2008). Nutrient inputs of N and P into aquatic systems originate from both point and nonpoint sources. Point sources are primarily associated with sewage effluents, including wastewater from households and industrial activities. Nonpoint or 'diffuse' sources include nutrients entering surface water systems in a dispersed manner, primarily linked to soil erosion from agricultural land use and the disturbance of

natural ecosystems, particularly through atmospheric N deposition (Sánchez-Colón and Schaffner, 2021). Over the past few decades (1980–2015), natural nutrient sources have been declining, and this trend is expected to continue through the coming decades (2015–2050) due to extensive land transformations, while nutrient inputs from agriculture, human sewage, and aquaculture are increasingly dominating (Beusen et al., 2022).

Despite its recognized importance, knowledge of water quality is hindered by significant gaps in the availability

and accessibility of in-situ data (Jones et al., 2024). For instance, 71% of water quality data is concentrated in North America and Western Europe, with major data gaps in regions such as the Middle East and North Africa, Southern Asia, and Eastern Europe and Central Asia (Jones et al., 2024). In the absence of sufficient in-situ data, process-based models (Jones et al., 2023), data-driven approaches (e.g., machine learning and artificial intelligence algorithms) (Zhu et al., 2022), and remote sensing (Nakkazi et al., 2024) are increasingly used to bridge this gap.

Process-based models are particularly effective in assessing the interactions between various processes in river basins, enhancing our understanding of how human-induced disturbances such as climate change, land use change, nutrient loading, and hydrological regulation affect pollutant levels in freshwater systems (Beusen et al., 2015). This makes physically based model approaches valuable for simulating water quality in ungauged catchments and projecting future water quality under changing climatic and socio-economic conditions (Wanders et al., 2019;

Srinivasan et al., 2010).



At the global scale, nutrient water quality models such as MARINA (Model to Assess River Inputs of pollutaNts to seAs; Strokal et al., 2021), IMAGE-GNM (Integrated Model to Assess the Global Environment–Global Nutrient Model; Beusen et al., 2022), and Global NEWS-2 (Nutrient Export from WaterSheds; Mayorga et al., 2010) have been applied. However, these global nutrient models typically have coarse spatial resolutions of 0.5 degrees (approx. 55 km at the Equator) and are generally run at multi-year time steps. In contrast, the Soil and Water Assessment Tool (SWAT+ model; Arnold et al., 1998; Bieger et al., 2017) offers several advantages, including the ability to operate at high spatial resolution using hydrological response units (HRUs) and simulate daily time steps. As an open-source tool, SWAT has seen widespread water quality application across various scales, from catchment to continental levels, in regions such as Europe (Abbaspour et al., 2015) and Africa (Nkwasa et al., 2024). Despite being data-intensive, SWAT(+) models can now benefit from high-performance computing (HPC) platforms, enabling the development of large-scale, high-resolution models that can be run and calibrated using parallel processing.

Here we present a global nutrient model (CoSWAT-WQ) set up using the SWAT+ modelling framework, which can simulate Total Nitrogen (TN) and Total Phosphorus (TN). The model is run at a river level globally and simulations are available at resolutions as fine as 2 km. Building on the work of Nkwasa et al. (2024) in Africa, we extend this modelling framework to a global scale, incorporating a range of hydro-climatic drivers and accounting for major point and non-point pollution sources. The high-spatio-temporal-resolution of CoSWAT-WQ, allows the model to provide globally consistent river nutrient assessments that are needed to identify hotspots and trends, especially in regions where in-situ data is insufficient for a detailed assessment. Considering that the SWAT+ model has been widely applied at local basin scale, by upscaling the modelling framework to a global scale while retaining basin/local-scale mechanistic understanding and governing processes, we hope that the COSWAT-WQ model can make a significant step towards bridging global, basin and local-scale water quality. As argued by Tang et al. (2019), mechanistic understanding from basin/local-scale models is essential for developing next-generation global water quality models with flexible scaling frameworks, allowing for the consideration of non-linearity and interactions across scales.

## 2. Materials and Methods

### 2.1 SWAT+ model description

SWAT+ is an enhanced version of the SWAT model, offering greater flexibility in linking spatial units and representing management practices (Bieger et al., 2017; Arnold et al., 2018). This semi-distributed model simulates hydrological processes, crop growth, sediment transport, and nutrient loading. It divides hydrological basins into sub-basins and Hydrologic Response Units (HRUs), where each HRU corresponds to areas with similar land use, soil type, slope, and management practices (Neitsch et al., 2005). Additionally, SWAT+ distinguishes upland and lowland processes through the use of landscape units (LSUs). The model's core functionality is based on the hydrological water balance, which drives all hydrological processes.



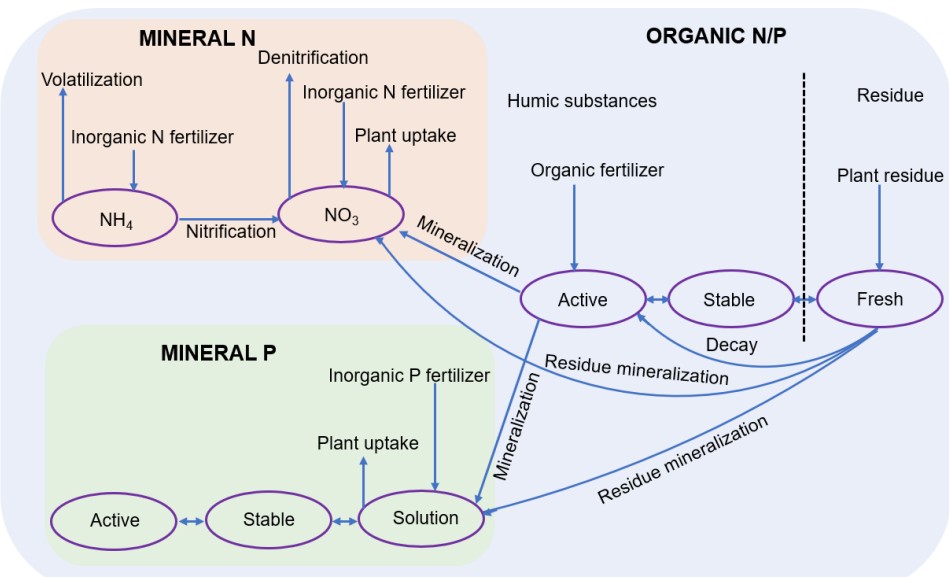

**Fig. 1**: Nitrogen (N) and phosphorus (P) transformation simulated in SWAT+ (Nkwasa, 2023)

SWAT+ estimates sediment yield for each HRU using the Modified Universal Soil Loss Equation (MUSLE), which
improves the prediction accuracy by incorporating surface runoff and peak flow rate data (Neitsch et al., 2005).
The model's channel sediment routing equation is derived from Bagnold's sediment transport equation (Bagnold,
1977), which accounts for sediment load entering the channel and either depositing excess sediment or re-
entraining sediment through channel erosion. Plant growth is simulated at the HRU level using a simplified version
of the EPIC growth model (Neitsch et al., 2005). Management practices that affect the plant growth cycle, such as
the timing of fertilizer and manure applications, irrigation, and biomass removal, can be scheduled based on
calendar days or heat units (Nkwasa et al., 2022a).

The fate and transport of nutrients (N and P) in a basin depend on nutrient cycling in the soil environment (in-
field) and stream channels (in-stream). The model simulates nitrogen (N) and phosphorus (P) cycles, tracking
various inorganic and organic forms of both nutrients (Fig. 1) in nutrient pools. N and P could increase or decrease
depending on their transformation and/or additions/losses occurring within each pool. Key processes in both cycles
include mineralization, decomposition, and immobilization. Nitrogen migration and transformation are calculated
based on water volume and soil nitrate concentration, while organic N and P transport with sediment is predicted
using topsoil concentrations, sediment yield, and an enrichment ratio. Nitrogen can be introduced into the soil
through several processes, including fertilizer and manure application, atmospheric deposition, and nitrogen
fixation. It can be removed via plant uptake, ammonia volatilization, denitrification, percolation, tile flow, lateral
flow, and surface runoff (Fig. 2). Phosphorus, on the other hand, is added primarily through fertilizer and manure
application and is removed through plant uptake, tile flow, lateral flow, percolation, and surface runoff (Fig. 2).

In-stream nutrient dynamics are modelled using the kinetic routines from the QUAL2E water quality model
(Brown and Barnwell, 1987). The model tracks both dissolved nutrients in the water and nutrients adsorbed to the



sediment. Dissolved nutrients move with the water, while those attached to sediments are allowed to settle and be
deposited on the channel bed. For additional information on nutrient processes in the SWAT+ model, refer to the
SWAT documentation (Arnold et al., 1998; Neitsch et al., 2005)

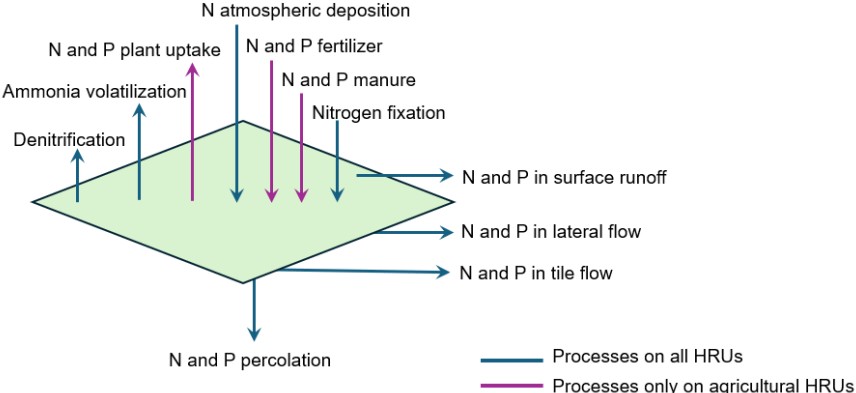

**Fig. 2**: Nutrient balance (N and P) processes on agricultural and non-agricultural HRUs

SWAT+ accounts for the groundwater component of the hydrological system through its aquifer module. Nitrate
entering the shallow aquifer via percolation from the soil profile can either remain in the aquifer, be transported to
the main channel via groundwater flow, move back into the soil zone in response to water deficits, or be transferred
to the deep aquifer through recharge (Neitsch et al., 2005). Due to the low mobility of phosphorus, SWAT+ permits
the leaching of soluble P only from the uppermost 10 mm of soil into the first soil layer. Although groundwater

flow from the shallow aquifer can transport soluble P to the main channel, the current version of SWAT+ does not
explicitly simulate this process.

### 2.2 Model setup

The modelling process, utilizing freely available global datasets as presented in Table 1, was organized as follows:
Initially, the default model was set up globally within the CoSWAT Workflow and run from 1995 to 2016, with a

focus on the hydrological component (Chawanda et al., 2025). A preliminary evaluation of the hydrological model
results, specifically river flow and evapotranspiration (ET), was conducted in Chawanda et al. (2025). For details
on the default hydrological model setup, which excludes crop management and pollution sources, please refer to
Chawanda et al. (2025). In this study, we extend the default hydrological model by incorporating pollution sources
(both point and non-point), as described in subsequent sections. The model is organized modularly, with the globe

divided into 90 large basins, each containing multiple sub-basins and hydrological response units (HRUs). These
large basins are run in parallel, and the results are aggregated at the global scale.

To add pollution sources, we utilize decision tables (Nkwasa et al., 2020) in our work. Decision tables enable the
user to model intricate sets of rules and their subsequent actions by allowing users to add conditions for scheduling
management (Arnold et al., 2018). For the nonpoint sources of pollution, the model applies workflows similar to

(Nkwasa et al., 2022a; Nkwasa et al., 2022b), where crop phenology (plant and harvest days) and crop management
(irrigation, N and P fertilizer and manure application rates) was extracted from the global datasets for both rainfed



and irrigated areas (HRUs), while N atmospheric deposition was extracted for all HRUs globally (Fig.3). The point source (PS) input data for each subbasin was sourced from Beusen et al. (2022). However, this data was provided as TN and TP loads, whereas SWAT requires inputs in different forms of nitrogen and phosphorus. To address this, ratios were defined for organically bound nitrogen to inorganic NH3 and NH4 at 0.25:0.75, and for organic phosphate to orthophosphate at 0.25:0.75, based on previous studies (Rossle and Pretorius, 2001; Hoxha et al., 2022). These ratios are conservative and will be updated in future versions, with spatial adjustments made based on regional or continental differences. For example, regions such as Europe and North America generally have higher treatment rates, whereas Africa tends to have lower treatment rates.

**Table 1**. Global datasets used for the modelling process

| Global Datasets | Resolution | | Source |
|---|---|---|---|
| | Temporal | Spatial | |
| Digital Elevation Model (DEM) | - | 2000 m | Advanced Spaceborne Thermal Emission and Reflection Radiometer (ASTER; Tachikawa et al., 2011) |
| Land use | - | 2000 m | European Space Agency (ESA; Defourny et al., 2012) |
| Soil | - | 2000 m | FAO–UNESCO global soil map |
| Climate (precipitation, temperature, windspeed, solar radiation and relative humidity) | Daily | 0.5º | EartH2Observe, WFDEI and ERA-Interim data Merged and Bias-corrected for ISIMIP (EWEMBI; Lange, 2016) |
| Irrigated areas | - | 0.083º | Food and Agriculture Organization (FAO; Siebert et al., 2013) |
| Plant and harvest dates | - | 0.5º | Global Gridded Crop Model Intercomparison (GGCMI; Jägermeyr et al., 2021) |
| Fertilizer use rate (N and P) | Yearly | 0.5º | (Lu and Tian, 2017; Hurtt et al., 2020; Beusen et al., 2022) |
| Manure use rate (N and P) | Yearly | 0.5º | (Potter et al., 2010) |
| Atmospheric N deposition | Monthly | 0.5º | ISIMIP (https://www.isimip.org; Tian et al., 2018) |
| Point source (wastewater discharge) | 5 -year | 0.5º | (Beusen et al., 2022) |
| Observed river TN and TP concentrations | Daily | River gauge | Global Freshwater Quality Database (GEMStat; https://gemstat.org/) |

**2.3 Model evaluation**

The CoSWAT-WQ model, like most global water quality models such as DynQual v1.0 (Jones et al., 2023), MARINA (Strokal et al., 2016), and IMAGE-GNM (Beusen et al., 2015) is uncalibrated. This is primarily due to





the lack of sufficient observational data, strong spatial biases in available observations, uncertainties in input
datasets, and the challenges of managing large sets of calibration parameters (Jones et al., 2023; Jones et al., 2024).
Furthermore, global calibration poses a significant challenge because many regions facing critical water quality
and quantity issues, such as Africa and parts of Asia, often lack adequate observational and validation data (Burek
et al., 2020). However, SWAT was designed to predict the impacts of management practices on water, sediment,
and agricultural yields in large, ungauged basins (Arnold et al., 1998; Srinivasan et al., 2010), making it
theoretically applicable in such regions without a significant loss in performance. Furthermore, uncalibrated
physical models are often preferred for global change assessments, particularly when evaluating a range of climatic
and socio-economic scenarios (Wanders et al., 2019).

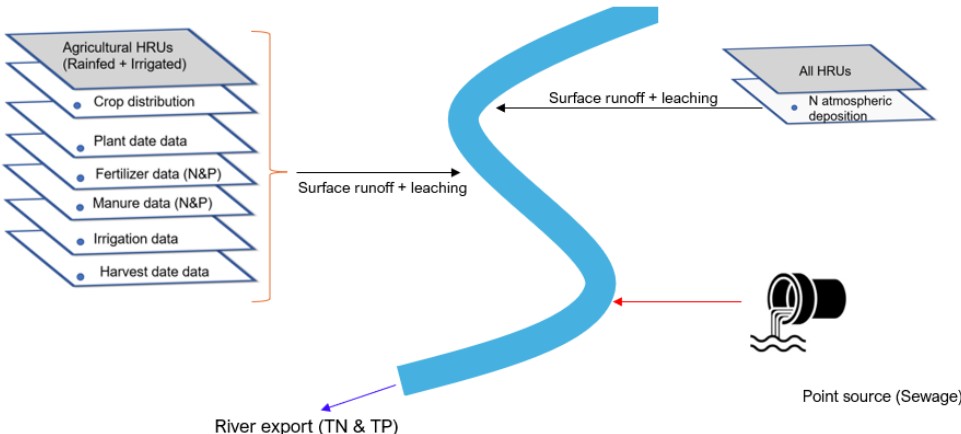

**Fig. 3**: Point & Non-point sources application scheme using decision tables

Thus, model evaluation efforts for this initial model version focused on comparing river nutrient loads with
estimates from other global nutrient models. Model intercomparison is a strategy proposed by Strokal et al. (2024)
to enhance confidence in large-scale water quality models. Additionally, river concentration simulations were
compared to global surface water quality monitoring station observations, with monthly temporal resolution for
the period 1995–2016. Observed data were sourced from the Global Environment Monitoring System for
Freshwater (GEMS/Water) database (GEMStat), a prominent repository of in-situ freshwater quality
measurements. However, the observational data are fragmented temporally and unevenly distributed spatially, with
a marked bias toward North America and Western Europe for TN and TP river concentrations. Due to the
limitations in the available observational records, global water quality models have generally not been evaluated
using conventional metrics applied in hydrological modelling, such as coefficients of determination, Nash–
Sutcliffe efficiency (NSE), and Percent Bias (PBIAS) (Jones et al., 2023; Beusen et al., 2015; Strokal et al., 2016).
Consequently, we adopted a model evaluation approach similar to those used in other global water quality
modelling efforts, employing statistical metrics such as normalized root mean square error (nRMSE) (Beusen et
al., 2015) and Kling–Gupta efficiency (KGE) (Jones et al., 2023). To assess the temporal performance of the model,
we present monthly simulation results for the two stations with the most extensive data availability for TN and TP



globally during the simulation period (Fig. 8). Additional model plots of simulations against observations are provided in the supplementary material A (section S1 – S2).

### 3. Results

#### 3.1 Global TN and TP load export

We compared simulated riverine exports of total nitrogen (TN) and total phosphorus (TP) in the world's 30 largest
rivers (Fig.4) using the CoSWAT-WQ model and the IMAGE-GNM model (Beusen et al., 2022). The analysis revealed strong agreement between the two models, with $R^2$ values of 0.87 for TN and 0.71 for TP (Fig. 5). We selected the IMAGE-GNM model for comparison because it is the only other global nutrient model that is both spatially explicit and operates at a grid scale. To address differences in model resolution, we focused on larger rivers. The IMAGE-GNM model runs at a coarser 0.5-degree resolution (~55 km), whereas CoSWAT-WQ operates
at a finer 2 km resolution. This difference in spatial resolution likely contributes to the observed variations in nutrient load estimates between the two models.

Our analysis of annual average TN and TP export loads to coastal waters for the period 1995–2016 (Fig. 4a – 4b) yielded results that are consistent with prior studies. The CoSWAT-WQ model estimated global TN export at 42.3 Tg/yr, which aligns closely with the lower range of published estimates, such as 39.1 Tg/yr from IMAGE-GNM
and 34.8 Tg/yr from the MARINA model (Strokal et al., 2021). Similarly, CoSWAT-WQ estimated global TP export at 5.3 Tg/yr, which is in agreement with IMAGE-GNM's 4.3 Tg/yr and MARINA's 2.9 Tg/yr load ranges. It is important to note that the MARINA model only simulates dissolved nitrogen and phosphorus, which may partly explain its lower export estimates compared to the CoSWAT-WQ model.





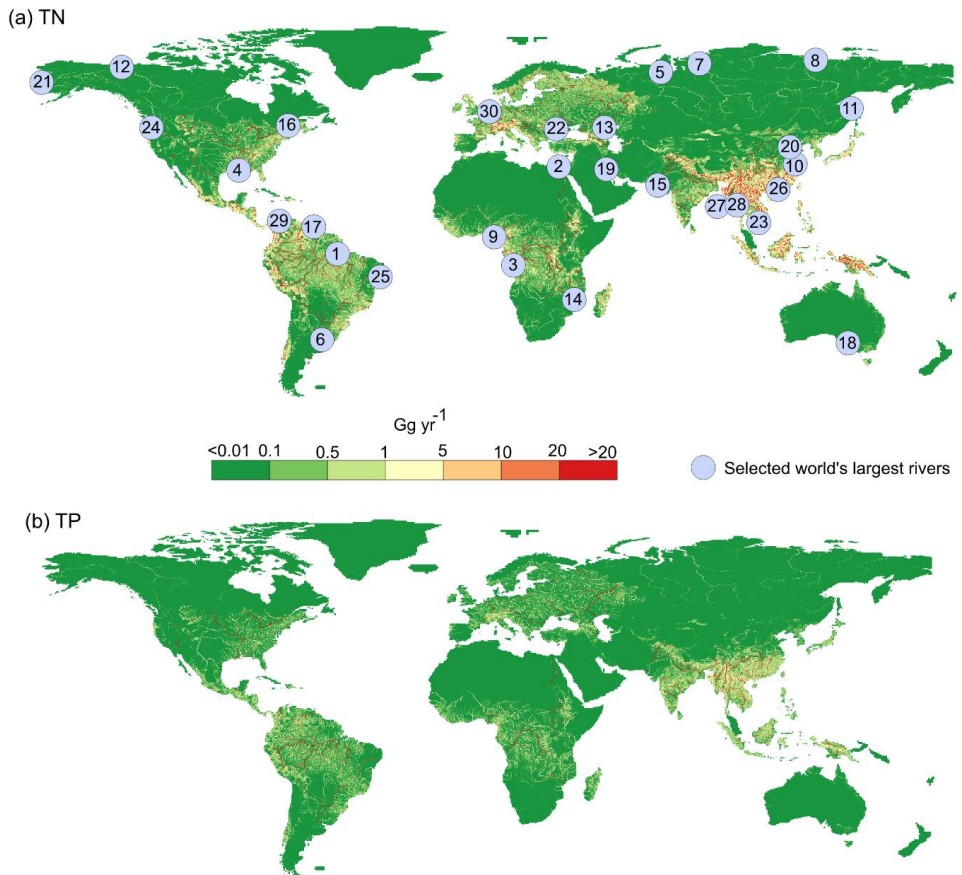

**Fig. 4**: Average annual (a) TN and TP river loading to coastal zones from the world's largest rivers during the period 1995–2016. The largest rivers are defined here as those with a basin area exceeding 0.164 million km², an average annual water discharge above 2,400 m³/s, and a length greater than 1,400 km (Best, 2019), numbered as 1 – Amazon, 2 – Nile, 3 – Congo, 4 – Mississippi, 5 – Ob, 6 – Parana, 7 – Yenisey, 8 – Lena, 9 – Niger, 10 – Yangtze, 11 – Amur, 12 – Mackenzie, 13 – Volga, 14 – Zambezi, 15 – Indus, 16 – St Lawrence, 17 – Orinoco, 18 – Murray-Darling, 19 – Shatt el Arab, 20 – Yellow, 21 – Yukon, 22 – Danube, 23 – Mekong, 24 – Columbia, 25 – Sao Francisco, 26 – Pearl, 27 – Irrawaddy, 28 – Salween, 29 – Magdalena and 30 – Rhine





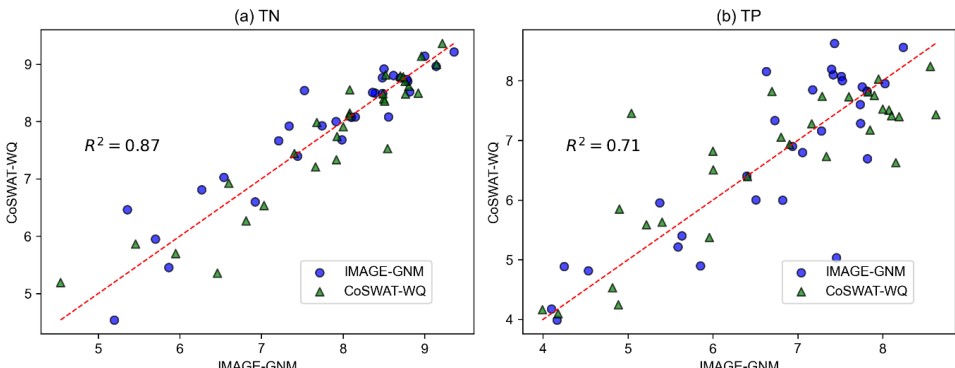

**Fig. 5**: CoSWAT-WQ vs IMAGE-GNM (log–log) for nutrient loadings from the selected world's largest rivers

**3.2 Spatial and temporal validation of TN and TP concentrations**

The statistical evaluation of the TN and TP simulations using the normalized Root Mean Square Error (nRMSE) and Kling-Gupta Efficiency (KGE) coefficient shows mixed results (Fig. 6 – 7). For nRMSE (Fig. 6), the model performs reasonably well, with over 80% of stations showing nRMSE values < 1 globally for both TN and TP river concentration simulations. However, the KGE results are less favourable, with only 53% and 49% of stations exceeding the KGE threshold of > −0.41 (Knoben et al., 2019) for TN and TP, respectively (Fig. 7). This

discrepancy can be explained by the model's tendency to underestimate (or bias) the observed values, even though it aligns with the observed data range (Fig. 8). As a result, while the nRMSE may appear low, KGE penalizes the model for this bias due to the mismatch between predicted and observed means. The low KGE performance at some stations may also be linked to the model's poor hydrological performance (Chawanda et al., 2025), as nutrient concentration is influenced by both river flow and nutrient load. Therefore, hydrological uncertainties likely affect

the river concentration predictions.

Additionally, for stations with more extensive observational data, the model performs better for TN (Fig 8(a) – (b)) than for TP (Fig 8(c) – (d)) concentrations. The model tends to underestimate the concentrations relative to the observed data, but the incomplete nature of the observational records complicates the assessment of whether this underestimation is consistent or if the observed concentrations are plausible. However, some stations show

good seasonality patterns, with the model capturing nutrient concentration peaks for both TN and TP (Supplementary material A, section S1 – S2). Despite the limitations in KGE performance, the model does capture the overall magnitude and seasonality of the observations within plausible ranges.



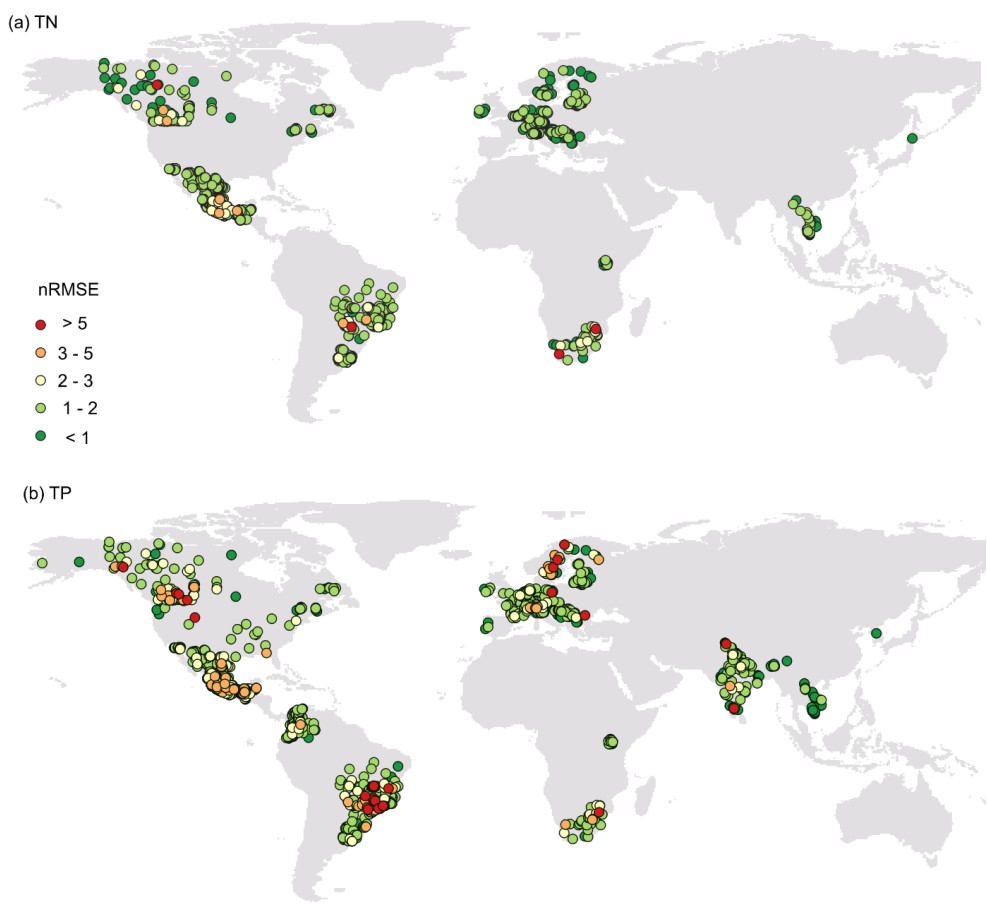

**Fig. 6**: Evaluation of model performance using normalized root mean square error (nRMSE) for (a) TN and (b) TP



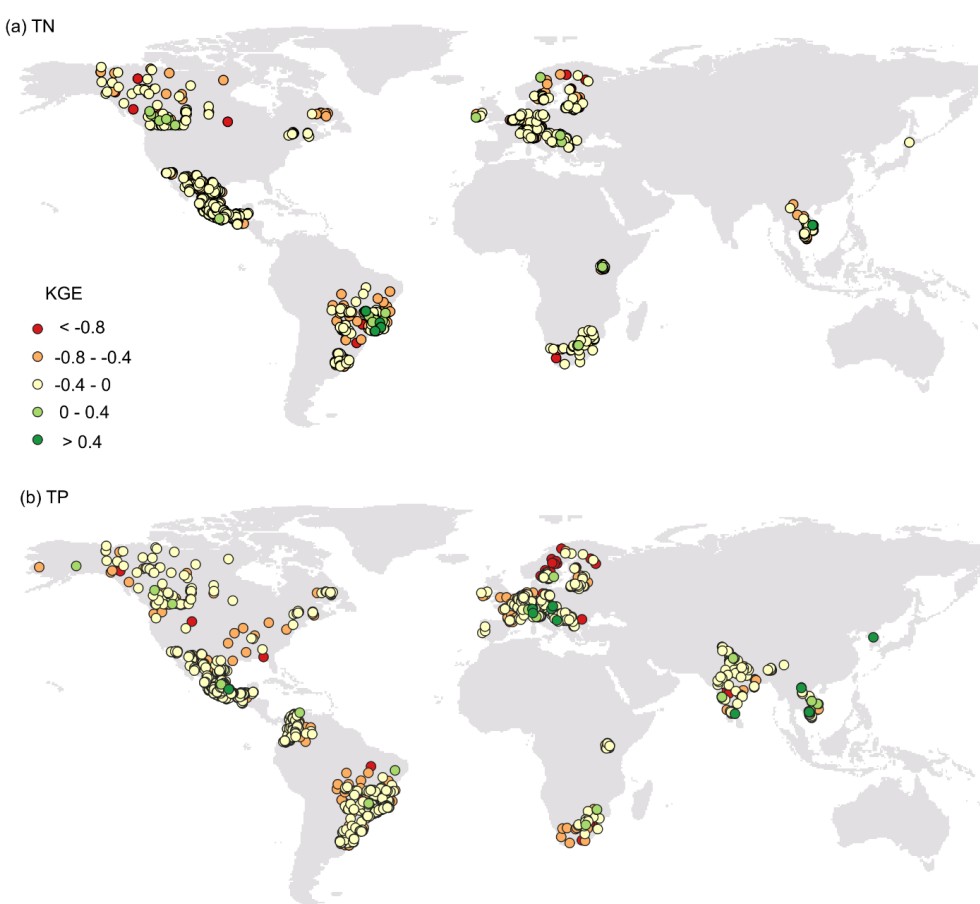

**Fig. 7**: Evaluation of model performance using Kling–Gupta efficiency (KGE) coefficient for (a) TN and (b) TP





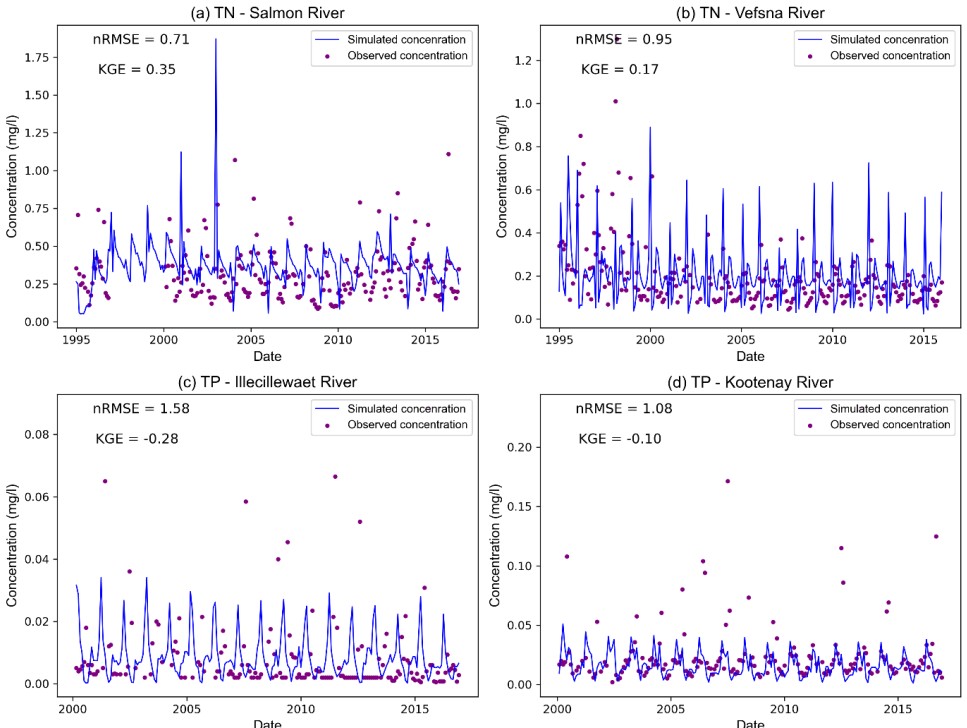

**Fig. 8**: Time series for observed and simulated river concentrations for TN (a – b) and TP (c – d) for the two stations with the most data availability for each constituent

## 4. Discussion

### 4. 1 Model validation

Based on comparisons with observational data and other global nutrient models, we can identify which parts of the model can be improved. However, it is difficult to determine specific areas for improvement given the uncertainties in the input data (e.g., land use, climate, hydrology, wastewater flows) and the complexity of surface, subsurface, and instream processes. Hydrology plays a particularly important role in model results, though this

discussion focuses on nutrient-related processes. Particularly, there is a need to improve river flow modelling and to incorporate water management features such as lakes and reservoirs, as highlighted in Chawanda et al. (2025). The SWAT model also allows for the calibration of key parameters affecting nutrient balance, such as plant uptake of nitrogen (N) and phosphorus (P), N and P percolation coefficients, the phosphorus availability index, and the organic N enrichment ratio. However, to better capture nutrient balances, reliable data on nutrient balance

components is necessary. Given the approximations inherent in model structure, uncertainties in input data, and the complexity of nutrient transport and transformation dynamics, the primary objective of global water quality models is not to predict exact daily concentrations (UNEP, 2016). Instead, they aim to identify major spatial and temporal pollution hotspots within global river networks. For this purpose, the CoSWAT-WQ model performs adequately in comparison to other global nutrient models for river nutrient loads but with room for improvement



in comparison to the concentration observation data. Future research will include further model validation and calibration, utilizing available observational data, particularly from more data-rich regions, to improve accuracy and reliability.

**4.2 Current missing links and future directions**

The current CoSWAT-WQ model has several limitations that need to be addressed to improve its representation of
nutrient pollution sources. For example, aquaculture, which has been increasingly identified as a significant source of diffuse pollution (Beusen et al., 2022), is not yet incorporated into the model. Furthermore, while manure application is currently limited to agricultural areas, it is also deposited in natural landscapes and grasslands, which is not accounted for in the present version. Additionally, the model does not include open defecation, despite studies showing that it contributes over 20% of nitrogen (N) and phosphorus (P) loads in some regions of Africa
and Southeast Asia (Strokal et al., 2021). Another gap is the weathering process, which, although a minor source of N and P compared to atmospheric deposition, biological fixation, and human-induced inputs, could improve the model's process representation. Future iterations of the model will aim not only to incorporate these processes to better capture the diverse sources of nutrient pollution but also to improve processes such as lakes and reservoir implementations as they have a big influence on nutrient enrichment and residence times.

Another future direction is the ability of the CoSWAT-WQ model to provide the future projections of multiple nutrients and forms at the global scale in a spatially consistent manner. Currently, the first version of the future projections of river nutrient load export and river concentrations from the CoSWAT-WQ model have been submitted to the Inter-Sectoral Impact Model Intercomparison Project - ISIMIP (https://www.isimip.org/) water quality sector. This will facilitate the intercomparison with other large scale models, helping identify areas of
agreements/disagreements. These insights can guide model improvements while also generating datasets that will facilitate the understanding of regional implications of anthropogenic increases in nutrient inputs and climate change.

The integration of the CoSWAT-WQ model with other sectoral models presents another promising avenue for future research. For example, nutrient loads simulated by CoSWAT-WQ can serve as inputs for lake models, such
as GPLake-M (van Wijk et al., 2023), enabling assessments of eutrophication levels in lakes. This linkage could guide policy development and management strategies aimed at mitigating lake eutrophication. Similarly, incorporating aquatic biodiversity models could allow for an evaluation of how varying nutrient concentrations affect aquatic species, thus supporting ecosystem-based management approaches (Nakkazi et al., in submission). Beyond nutrient pollution, the SWAT model offers the capacity to simulate additional water quality constituents,
such as suspended sediment load, water temperature, chlorophyll-a, biochemical oxygen demand (BOD), and salinity. Future versions of the CoSWAT-WQ model could leverage this capability to simulate a broader range of pollutants at the global scale.

While the current model primarily uses global management datasets, there is potential to enhance local management processes for specific sub-basins where detailed information is available. As it stands, the CoSWAT-
WQ model provides valuable insights that only large-scale water quality modelling can offer, with significant potential for future advancements in both model scope and accuracy.

**5. Conclusion**



We present the CoSWAT-WQ model that has been specifically developed to obtain patterns of nutrient export from rivers at regional to global scales at different resolutions with different datasets. The model's ambition lies in its ability to provide consistent global surface water quality simulations at high spatio-temporal resolutions (i.e., 2 km and daily time steps), while offering a unique framework for bridging water quality assessments at global, basin, and local levels. The performance of our global nutrient model is similar to that of other global nutrient models and performs relatively well without calibration, especially with nRMSE coefficients. However, it is important to acknowledge the challenges posed by observational data, including variability and potential inaccuracies in reporting, which may influence model performance (Jones et al., 2023; Supplementary material A, section S3).

CoSWAT-WQ is freely accessible and highly customizable, allowing users to tailor input and output data according to their specific research needs. This flexibility enables a wide range of applications, supporting investigations at scales from global to local. The model can facilitate diverse research inquiries, including the attribution of nutrient export to climate change, the evaluation of agricultural policies or management strategies on nutrient export, and the assessment of the impacts of future socio-economic developments and climate extremes on river nutrient dynamics. In this way, CoSWAT-WQ is positioned to serve as a valuable tool for scientists, policymakers, and other stakeholders engaged in water quality research and management.

**Code and data availability**

CoSWAT-WQ v1.0 is an open-source model currently hosted on the VUB-HPC (High-Performance Computing; https://hpc.vub.be/) cluster due to the large model file size (over 8 TB). The development and maintenance of CoSWAT-WQ v1.0 is managed by the Department of Water and Climate at Vrije Universiteit Brussel (VUB) and the Water Security Group at the International Institute for Applied Systems Analysis (IIASA). However, contributions from external parties are welcome and encouraged.

The codes used in this study are available open-access through a GitHub repository (https://github.com/VUB-HYDR/2024_Nkwasa_et_al) and the simulated data for TN and TP loads and concentrations covering the period of 1995 – 2016, has been made available on zenodo (https://zenodo.org/records/14260298). A sample model set-up with all source files and output files for the Irrawaddy river basin is provided as an example (https://zenodo.org/records/14264507).

**Acknowledgements**

The authors thank the Research Foundation – Flanders (FWO) for funding the International Coordination Action (ICA) "Open Water Network: Open Data and Software tools for water resources management" (project code  the Open Water Network: G0E2621N), impacts of global change on water quality (project code G0ADS24N), the AXA Research Chair fund on Water Quality and Global change. The computational resources and services used in this work for the simulations and storage of CoSWAT-WQ v1.0 model data were provided by the VSC (Flemish Supercomputer Center), funded by the Research Foundation - Flanders (FWO).

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
