# Peer review of "CoSWAT-WQ v1.0: a high-resolution community global SWAT+ water quality model"

_EGUsphere, 2025_

## Author Comment (AC1)

**Note:** The comments from the reviewer are in black while the responses from the authors are in blue.

**Reviewer 1**

The manuscript presents the new CoSWAT-WQ model, which enhances our understanding of global water quality problems related to nutrients, particularly through its high spatial and temporal resolution. This represents an important step in the field of water quality research and paves the way for future model developments. I read the manuscript with great interest. It is generally well-written and presents a new global water quality model with a remarkably high resolution. This model, indeed, holds potential for scientists, policymakers and other stakeholders in the field of water quality. However, currently, the limited methodological detail makes it challenging to fully assess the scientific approach and methods applied. Moreover, the manuscript could highlight the novel insights gained from the model results rather than solely focusing on model evaluation. Improvements in the Methods, Results, and Discussion sections are needed before considering this manuscript for publication. I have explained my concerns in the comments below.

**Response:** We sincerely thank the reviewer for their thoughtful and encouraging comments, as well as their interest in our work. We appreciate the recognition of the potential and significance of the CoSWAT-WQ model for advancing global water quality research. We acknowledge the concerns raised and will incorporate all suggestions to improve the quality and clarity of the manuscript.

1. One of the main novelties of the new CoSWAT-WQ model is its high spatial and temporal resolution at the global scale. Yet, this also comes with some concerns:

   o The authors present the model as a daily time step model. Yet, the discussion includes the following: "Given the approximations inherent in model structure, uncertainties in input data, and the complexity of nutrient transport and transformation dynamics, the primary objective of global water quality models is not to predict exact daily concentrations (UNEP, 2016). Instead, they aim to identify major spatial and temporal pollution hotspots within global river networks. For this purpose, the CoSWAT-WQ model performs adequately in comparison to other global nutrient models for river nutrient loads but with room for improvement in comparison to the concentration observation data". I would suggest the authors clarify their justification for a daily time-step model and the implications of this choice, especially at the global scale. This particularly holds for tropical and subtropical regions where scheduling using heat units may result in incorrect cropping seasons, as noted in Nkwasa et al., (2022). This raises questions such as "how well does the model capture daily patterns?" and "how should the reader interpret the model outputs?" Another option is to revise the aim of the study and clearly link the purpose of the model and how this fits the daily time step.

**Response:** Thank you for raising this point. We recognize that the current description may have caused a lack of clarity regarding the model's temporal focus and subsequently, the aim of the study. Although the CoSWAT-WQ is run at a daily timestep, our evaluation has most been done at monthly (with observations) and annual timestep (with other global models). Thus, recommending the current model version for identifying major spatial and

temporal pollution hotspots rather than predicting exact daily concentrations. We recognize that this position may appear contradictory to statements in the abstract and conclusion about the model's daily simulation capability. To clarify, we will revise the manuscript to explicitly state that although the model runs on a daily timestep, its current validated use is for broader temporal patterns monthly and annual and spatial hotspot identification. At the same time, we emphasize that the model's daily timestep structure offers significant potential for future improvements, with subsequent versions leveraging daily simulations to provide higher-resolution estimates supported by high-resolution validation.

Regarding cropping season scheduling and concerns about heat unit-based approaches Nkwasa et al. (2022), we actually follow the workflows from Nkwasa et al. (2022) to implement management practices, using global cropping calendars based on actual planting and harvest dates rather than relying on heat units. This approach helps ensure more accurate representation of cropping seasons, particularly in tropical and subtropical regions.

*We will revise the manuscript to better align the study's aims with the model's current strengths in hotspot identification and highlight the potential of the daily timestep for future developments.*

- o I acknowledge that model evaluations at the global scale are challenging. Hence, I appreciate the authors efforts in evaluating the model by comparing the model results with the IMAGE-GNM model (Figures 4-5) and monitoring data from GEMStat (Figure 8). For the comparison with IMAGE-GNM, the authors compared simulations for the world's 30 largest rivers. To enhance the relevance of this comparison, I suggest including information such as the percentage of the global total drainage area that these rivers cover or their share related to total nutrient exports. In Figure 5, it is difficult to interpret the different symbols used (circle for IMAGE-GNM and triangle for CoSWAT-WQ). The use of symbols seems unnecessary as each model is included on a different axis. In addition, I wonder whether the co-authors have considered comparing their model results to other large-scale water quality models (e.g. mQM for Europe and SWAT+ for Africa). Regarding the comparison to monitoring data from GEMStat (the temporal patterns in Figure 8), it becomes clear that the model may underestimate extreme events and shows a slightly advancing pattern (e.g. earlier peaks) compared to the monitoring data. Next to GEMStat, there are other databases available. For example, Jones et al. (2024) compiled a comprehensive dataset of water quality monitoring data. Including this dataset in the evaluation can further strengthen the evaluation efforts.

**Response**: Thank your for your suggestions. We will include the information such as the percentage of the global total drainage area that these rivers cover or their share related to total nutrient exports as shown in Table R1. Regarding the symbols in Figure 5, we believe that their inclusion helps readers distinguish where each model's data points lie in relation to the 1:1 line. Although the models are plotted on separate axes, the symbols add an extra layer of clarity for comparative interpretation.

With respect to comparisons to other large-scale or regional models, such as SWAT+ for Africa (Nkwasa et al., 2024) and mQM for Europe (Kumar et al., 2020), we agree that these are valuable references. We will include a spatial comparison with these models in the supplementary material to complement our global evaluation.

As for the suggestion to incorporate additional observational datasets beyond GEMStat, we agree that a broader evaluation can enhance robustness. However, GEMStat remains the most extensive individual publicly available global database for water quality. For example, the average time series length per site for total phosphorus (TP) in GEMStat is 6.6 years, compared to 4.9 years in GLORICH. Moreover, many of the sites in GEMStat overlap spatially with other databases such as Waterbase, the Water Quality Portal (WQP), and the Canadian Environmental Sustainability Indicators (CESI), often resulting in duplicate records (Virro et al., 2021). Importantly, large regions in Africa, South America, and Asia remain underrepresented in alternative datasets (Jones et al., 2024), limiting their utility for a truly global-scale evaluation. Given these considerations, we consider GEMStat a sufficient and appropriate choice for the scope of this study.

*We will revise the manuscript to enhance the inter-model comparison (including Table R1) and include an evaluation of our global model against regional models in Africa and Europe, both in the main text and the supplementary material.*

Table R1: Model intercomparison of CoSWAT-WQ, IMAGE-DGNM (Beusen et al., 2015) and MARINA-Multi (Micella et al., 2024) models for the year 2010

| River ID | River basin name | Area (km2) | mouth longitude | mouth latitude | TN (Tg/yr) | | | TP (Tg/yr) | | |
|---|---|---|---|---|---|---|---|---|---|---|
| | | | | | IMAGE-DGNM | MARINA-Multi | CoSWAT-WQ | IMAGE-DGNM | MARINA-Multi | CoSWAT-WQ |
| 1 | Amazon | 5846870 | -51.75 | -1.25 | 3.78 | 4.10 | 1.64 | 0.17 | 0.29 | 0.36 |
| 2 | Nile | 3821590 | 31.25 | 31.25 | 0.16 | 0.36 | 1.39 | 0.01 | 0.04 | 0.07 |
| 3 | Congo | 3694430 | 12.75 | -5.75 | 0.58 | 1.27 | 0.83 | 0.00 | 0.11 | 0.14 |
| 4 | Mississippi | 3199170 | -90.25 | 29.75 | 1.38 | 0.99 | 0.92 | 0.11 | 0.07 | 0.08 |
| 5 | Ob | 3022320 | 69.25 | 66.75 | 0.29 | 0.25 | 1.33 | 0.03 | 0.02 | 0.04 |
| 6 | Parana | 2660890 | -58.75 | -34.25 | 0.61 | 0.59 | 0.38 | 0.05 | 0.07 | 0.06 |
| 7 | Yenisey | 2575660 | 82.25 | 71.25 | 0.28 | 0.001 | 0.001 | 0.00001 | 0.02 | 0.002 |
| 8 | Lena | 2438900 | 127.25 | 73.25 | 0.24 | 0.001 | 0.001 | 0.0002 | 0.01 | 0.0001 |
| 9 | Niger | 2237360 | 6.75 | 4.75 | 0.38 | 0.39 | 0.10 | 0.004 | 0.10 | 0.05 |
| 10 | Yangtze | 1792120 | 121.75 | 31.25 | 1.13 | 1.92 | 1.22 | 0.01 | 0.14 | 0.27 |
| 11 | Amur | 1752600 | 140.75 | 53.25 | 0.42 | 0.19 | 1.77 | 0.03 | 0.02 | 0.17 |
| 12 | Mackenzie | 1692900 | -134.75 | 69.25 | 0.17 | 0.10 | 1.63 | 0.00 | 0.01 | 0.00 |
| 13 | Volga | 1474650 | 48.25 | 46.25 | 0.18 | 0.19 | 0.91 | 0.03 | 0.01 | 0.14 |
| 14 | Zambezi | 1361960 | 36.25 | -18.75 | 0.42 | 0.15 | 0.11 | 0.00 | 0.01 | 0.03 |
| 15 | Indus | 1141750 | 67.75 | 24.25 | 0.50 | 0.13 | 1.36 | 0.11 | 0.003 | 0.09 |
| 16 | St Lawrence | 1052470 | -70.75 | 47.25 | 0.01 | 0.16 | 0.02 | 0.03 | 0.02 | 0.002 |
| 17 | Orinoco | 1038130 | -61.25 | 8.75 | 0.30 | 0.95 | 0.42 | 0.03 | 0.09 | 0.08 |
| 18 | Murray-Darling | 1030290 | 139.25 | -35.25 | 0.02 | 0.04 | 0.001 | 0.003 | 0.001 | 0.002 |
| 19 | Shatt el Arab | 988948 | 48.25 | 30.25 | 0.10 | 0.004 | 0.08 | 0.02 | 0.002 | 0.02 |
| 20 | Yellow | 892570 | 118.25 | 37.75 | 0.65 | 0.08 | 0.00 | 0.07 | 0.02 | 0.00 |
| 21 | Yukon | 854690 | -164.75 | 62.75 | 0.04 | 0.04 | 0.23 | 0.01 | 0.004 | 0.009 |
| 22 | Danube | 787069 | 29.25 | 45.25 | 0.37 | 0.26 | 0.51 | 0.05 | 0.02 | 0.07 |
| 23 | Mekong | 757660 | 106.25 | 10.25 | 0.57 | 0.60 | 0.45 | 0.14 | 0.03 | 0.01 |
| 24 | Columbia | 731105 | -123.75 | 46.25 | 0.14 | 0.09 | 0.96 | 0.01 | 0.01 | 0.16 |
| 25 | Sao Francisco | 614418 | -36.75 | -10.25 | 0.10 | 0.13 | 0.08 | 0.01 | 0.01 | 0.02 |
| 26 | Pearl | 408043 | 113.25 | 22.25 | 0.36 | 0.82 | 0.01 | 0.03 | 0.05 | 0.002 |
| 27 | Irrawaddy | 405481 | 95.25 | 15.75 | 0.78 | 0.43 | 0.58 | 0.16 | 0.04 | 0.09 |
| 28 | Salween | 273038 | 97.25 | 16.75 | 0.26 | 0.16 | 0.01 | 0.08 | 0.01 | 0.09 |
| 29 | Magdalena | 251445 | -74.75 | 10.75 | 0.30 | 0.34 | 0.31 | 0.05 | 0.05 | 0.04 |
| 30 | Rhine | 164864 | 5.75 | 52.75 | 0.15 | 0.17 | 0.21 | 0.01 | 0.01 | 0.02 |

2.  The Method section seems to provide only a limited description of several important aspects of the model. Below, I outline specific areas where further detail would improve clarity and reproducibility.

   o   In Lines 114-116, the authors state that the timing of fertilizer and manure applications, as well as irrigation and biomass removal, can be scheduled based on calendar days or heat units, as described in Nkwasa et al. 2022. To fully understand the method, readers may need to consult previous SWAT model documentation. Given that this manuscript emphasizes the novelty of its spatial and temporal resolutions, it would be beneficial to include a brief overview of the downscaling procedures – either directly in the text or as part of a conceptual framework. For example, a brief description on how the timing of manure and fertilizer applications (which are annual datasets, as indicated in Table 1) are determined would enhance clarity. This is particularly important as the timing can be set by the user or is automatically applied by SWAT based on a specified nitrogen stress threshold (according to Neitsch et al., 2005). Additionally, it would be helpful to include a short description of how the 5-year point source input data from Beusen et al. (2022) were downscaled, as well as how the 0.5-degree resolution input data were spatially refined to 2 km resolutions (Table 1).

**Response:** Thank you your suggestions. We will add the details on how we apply fertilizer in the revised manuscript as this has also been raised in the Community Comment. To give some details here, consider an agricultural grid or hydrologic response unit (HRU), as shown in Figure R1. Above each of these grids, we have several management layers, including crop type, cropping calendar (plant and harvest days), and fertilizer or manure application. For each agricultural grid or HRU, we read the corresponding values from these management layers and apply them accordingly. This process is pre-processed using a Python script, which organizes the information into a decision table (an example is shown in Figure R2), following the workflow developed in Nkwasa et al. (2022). This allows us to automate the assignment of agricultural management practices across all agricultural grids or HRUs.

[Figure]

**Fig. R1**: Schematic representation of an agricultural grid or hydrologic response unit (HRU) and its associated management layers

Figure R2 presents a sample decision table covering planting, harvesting, and fertilizer application. It illustrates different nitrogen (N) and phosphorus (P) practices i.e. amounts, frequencies, and application method. The total annual fertilizer amounts come from existing global datasets; these annual totals are divided into three equal portions (application frequency) (Hu et al., 2021), that are broadcast (application type) after the planting date and

before the harvest date, spaced across the cropping season. This preserves the correct yearly total but can still introduce uncertainty, because actual farm practices vary worldwide in timing, frequency, and method of application. Thus, we do not use the nutrient (N and P) stress threshold to trigger fertilization but rather apply the annual fertilizer totals into 3 equal portions across the cropping seasons.

[Figure]

**Fig. R2**: Sample decision table showing the planting, harvesting and fertilizer application harvesting of a wheat crop

The question of downscaling the resolution of the datasets used for the management layers. In our study, we use freely globally available datasets, most of which are at a resolution of 0.5 degrees. Since our model operates at 2 km grid resolution, multiple 2 km grid cells fall within each 0.5-degree cell. This means that all agricultural grid cells within a single 0.5-degree grid share the same management practices. In summary, the degree of heterogeneity in management practices we can capture depends on the resolution of the agricultural input datasets. To our knowledge, there are currently no global management datasets available at a higher resolution than those we have used. However, one strength of our workflow is its flexibility, if higher-resolution data become available in the future, this approach can incorporate and reflect that added heterogeneity.

To downscale the 5-year point source input data from Beusen et al. (2022) to annual resolution, each 5-year data point was assumed to represent the average conditions for the surrounding 5-year period. Specifically, the value for a given year (e.g., 1990) was applied uniformly to the five-year span from 1990 to 1994. Similarly, the 1995 data point was used for the years 1995–1999, and so on. This step-wise temporal downscaling approach assumes constant point source inputs within each 5-year interval.

*We will revise the manuscript to include more detailed descriptions of the management workflows in both the main text and supplementary materials, particularly in relation to fertilizer application and point source data integration.*

- o Section 2.1 refers to the N and P cycles and associated nutrient pools. However, it remains unclear how the model ensures mass balance and closure of these nutrient pools across space (e.g. line 117 refers to basins whereas the model runs on HRU scale) and time (e.g. negative balances).

**Response:** Thank you for your question. To clarify, in SWAT+, nutrient cycling and mass balance calculations are performed independently at the HRU level, where processes such as plant uptake, leaching, and mineralization are

simulated daily. These nutrient fluxes from all HRUs within a subbasin are then aggregated by summing components like surface runoff nitrogen and leached nitrate and passed to the subbasin-level routing unit. Within the subbasin, SWAT+ models in-stream and lateral processes such as nutrient settling, decay, channel transport as nutrients move through the stream network. The resulting subbasin-scale nutrient balance reflects the cumulative contributions from all HRUs along with transformations occurring within the stream system. Mass balance is maintained by explicitly tracking all nutrient inputs, outputs, and transformations across both land and water components. Temporally, SWAT+ updates all nutrient pools on a daily time step and applies the mass balance framework: all inputs, outputs, and transformations are accounted for, ensuring that changes in storage reflect the difference between inflows and outflows. In SWAT+, negative nutrient pools are not directly simulated as physical entities. Instead, negative values in nutrient pool outputs often indicate model behavior where losses (e.g., through denitrification, leaching, or erosion) exceed inputs during a specific time step. A positive nutrient balance value means that the nutrient is being retained in the system and negative nutrient balance means that the nutrient inputs are lower than outputs during a given period, considering the outputs as the losses by all the possible pathways.

*We will revise the manuscript to include more details about the nutrient balance and the simulations from HRU to basin scale.*

> o   In contrast to the calibrated SWAT+ model, the new CoSWAT-WQ model is uncalibrated. Yet, it remains unclear which model parameters used to be calibrated and how the uncalibrated alternative works. I suggest the authors to include a table which specifies which model parameters are calibrated in SWAT+ and how the uncalibrated approach works.

**Response:** Thank you for your suggestion. The current version of the CoSWAT-WQ model, like most global water quality models such as DynQual v1.0 (Jones et al., 2023), MARINA (Micella et al., 2024), and IMAGE-GNM (Beusen et al., 2015) is uncalibrated. We used the default CoSWAT-WQ model parameters, which are based on literature and previous applications of the SWAT and SWAT+ models (Arnold et al., 2012; Arnold et al., 2013). As noted in the discussion (Line 270), future research will focus on model calibration and validation using observational data, especially from data-rich regions, to enhance the model's accuracy and reliability.

*We will include the table of the default model parameters (Table R2) in the Supplementary material of the revised manuscript to show which parameters are usually/typically calibrated for hydrology and nutrients.*

Table R2: Values of SWAT+ parameters that are typically subject to calibration for hydrology and nutrients

| Parameter | Description | Common calibration range |
|-----------|-------------|--------------------------|
| **Flow** | | |
| CN2 | Curve number (percent) | ±10% of default value |
| ESCO | Soil evaporation compensation coefficient (replace) | 0-1 |
| AWC | Soil available water capacity (relative) | ±0.04 of default value |
| perco | Percolation coefficient (replace) | 0-1 |
| Surlag | Surface runoff lag coefficient (relative) | 0.05-24 |
| Latq_co | Lateral flow coefficient (replace) | 0-1 |
| **Phosphorus** | | |
| P_updis | P uptake distribution parameter | 0-100 |
| Phoskd | Phosphorus soil partitioning coefficient | 100-400 |
| biomix | Biological mixing efficiency | 0-1 |
| psp | Phosphorus availability index | 0.01-0.7 |
| Lat_orgp | lateral organic phosphorus | 0-200 |
| **Nitrogen** | | |
| cdn | Denitrification exponential rate coefficient | 0-3 |
| cmn | Rate factor for humus mineralization | 0.001-0.003 |
| sdnco | Denitrification threshold water content | 0-1 |
| N_perco | Nitrate percolation coefficient | 0-1 |
| N_updis | N uptake distribution parameter | 0-100 |

The Results section provides a comparison of the model results with other models and monitoring data. However, it lacks a presentation of novel insights from the newly developed model. For example, what new understanding do the high-spatial-temporal-resolution results offer regarding global and local water quality assessments? From my point of view, highlighting such novelties would strengthen the value of the study and fit the aim of the study.

**Response:** We thank the reviewer for the valuable suggestion. We agree that the manuscript would benefit from a clearer articulation of the novel insights provided by the CoSWAT-WQ model. In the revised manuscript, we will strengthen the purpose of the study by emphasizing several key aspects e.g. (i) Although the current version of CoSWAT-WQ has not yet been validated at high temporal resolution, the model is capable of simulating intra-annual and even daily simulations in nutrient and pollutant loading. This temporal resolution provides the potential to better understand the timing of peak pollution events and align management actions accordingly, especially under global changes. (ii) By leveraging globally available datasets within a consistent modelling framework, CoSWAT-WQ supports scalable applications that can be used for both local catchment-level studies and global

water quality assessments. This dual capacity enhances its utility for comparative analysis across regions. (iii) We highlight the community-based nature of CoSWAT-WQ. Users can download and apply individual subbasin models for their own research, including further validation and calibration where local data is available. Importantly, users also have the option to contribute improved or calibrated parameter sets for specific subbasins back to the model database. This participatory structure is designed to enhance model accuracy over time and foster a collaborative global modelling effort.

*These points will be included in the revised manuscript to better convey the novelty and value of the CoSWAT-WQ model.*

3. The Discussion section could be strengthened by a more thorough examination of certain modeling choices and their implications on the study's findings.

   o Line 162, "The ratios are conservative and will be updated in future versions…". I suggest reflecting on the implications of the conservative ratios on the model outputs.

**Response:** We thank the reviewer for this suggestion. Applying the same speciation ratios for all subbasins globally introduces a degree of simplification that may not fully capture spatial variability in wastewater discharge characteristics. This may lead to over- or underestimation of both bioavailable (e.g., $NH_4^+$, $PO_4^{3-}$) and less bioavailable (e.g., organic N and P) nutrient fractions. For example, the model could underestimate bioavailable loads in regions with low treatment efficiency, or overestimate them where advanced treatment reduces reactive forms. Similarly, it could misrepresent the quantity of organic-bound forms that contribute to long-term nutrient cycling. While our current model version applies the same speciation fractions globally, future versions should incorporate region-specific speciation ratios to better capture spatial variability in nutrient forms as this would improve the accuracy of regional and local water quality assessments.

*These points will be included in the revised manuscript to better reflect on the speciation ratios/fractions.*

   o The study used and downscaled several global input datasets. In Section 4.1, the uncertainties in the input data are acknowledged in the Discussion section of the manuscript. Yet, the authors could elaborate more on the implications of these uncertainties on the results.

**Response:** We thank the reviewer for this comment. As noted, the model relies on several global input datasets, which inherently introduce uncertainties into our simulations. While these datasets are essential for achieving global coverage, they come with limitations that can affect model accuracy in multiple ways. For example, crop management data at a 0.5-degree resolution may fail to capture the heterogeneity of agricultural practices, including crop types, fertilizer application rates, and management intensity. This can lead to misrepresentation of nutrient source areas at the local level. Similarly, using gridded climate data (e.g., precipitation) at the same resolution can significantly influence hydrological processes such as runoff generation, which are key drivers of nutrient mobilization and transport. Point source nutrient loads, which are derived from global gridded datasets, also carry substantial uncertainty, especially because such data distribute emissions over grid cells, whereas in reality, wastewater discharges occur at specific point locations. This spatial generalization may lead to mismatches between modelled and actual nutrient input locations, particularly in urban or industrial regions. These issues are further compounded in regions with sparse observational data, where global datasets rely more heavily on

interpolation or modelling assumptions. As a result, model outputs in such areas may be less reliable in absolute terms, although they may still be useful for identifying relative patterns, trends, or hotspots. Thus, to improve the robustness of future simulations, regionally validated input datasets should be integrated wherever available.

*We intend to include this discussion in the revised manuscript.*

- o To my understanding, the SWAT+ model accounts for five crops that represent different croplands (Table 2 and Figure 2, Nkwasa et al., (2022)). However, globally, many different crops exist, with each having a characteristic cropping pattern. Hence, this raises the question of whether the representative crops as used in SWAT+ are also representative for a global application of the model (e.g. rice does not seem to be included). I suggest the authors justify their choice and reflect on uncertainties associated with the crop selection.

**Response:** We thank the reviewer for highlighting this important point. To give context, the global crop dataset we use classifies crops into broad functional types: C3 annual crops (C3ann), C3 perennial crops (C3per), C4 annual crops (C4ann), C4 perennial crops (C4per), and C3 nitrogen-fixing crops (C3nfx). For each cropland category, a single representative crop was selected based on the global crop distribution estimates by Leff et al. (2004), as shown in Table 2 (Nkwasa et al., 2022). For example, Both wheat and rice are classified as C3 annual crops; however, wheat was selected as the representative crop for this category because it occupies the largest (22 %) global cultivation area among the major crops (Leff et al., 2004),. However, we acknowledge that in certain regions, particularly South and Southeast Asia, rice is the dominant crop and plays a significant role in local hydrology and nutrient dynamics. This generalization introduces uncertainties in regional simulations, especially in areas where the selected representative crop differs significantly from the actual dominant crop in terms of phenology, rooting depth, water requirements, and nutrient uptake. In the case of rice, for example, flooded field conditions can lead to different nutrient transformation and transport processes compared to dryland wheat systems (Zhao et al., 2012). As a result, runoff and leaching estimates in rice-growing areas may be misrepresented in the current model setup. We recognize this as a limitation of using simplified, globally representative crop types for large-scale applications. Future model development will explore incorporating region-specific dominant crops, particularly in hotspot regions, to better reflect local agricultural practices and reduce uncertainty in nutrient loss estimates.

*We intend to include this discussion in the revised manuscript.*

- o Lines 282-284, "…but also to improve processes such as lakes and reservoir implementations as they have a big influence on nutrient enrichment and residence times". Could you provide some more insights on this? For example, include a reference or define 'a big influence'.

**Response:** We thank the review for this suggestion. To provide further context, lakes and reservoirs play a critical role in nutrient dynamics by through retention or transformation within a watershed. Globally reservoirs act as "sinks" for nitrogen and phosphorus, although the removal rate of phosphorus exceeds that of nitrogen (Gan et al., 2025). They influence nutrient enrichment and residence times through processes such as sedimentation, biological uptake, and nutrient recycling, which can significantly alter nutrient concentrations downstream. For example, reservoirs often increase water residence time, promoting nutrient settling and uptake by aquatic organisms, thereby reducing nutrient loads reaching downstream ecosystems (Yin et al., 2024). Similarly, lakes can retain

nutrients for extended periods, affecting timing and magnitude of nutrient export. Recent studies in the Amazon, the world's largest river basin have highlighted the substantial impact of damming, showing how individual tributary basins experience reductions in downstream sediment and nutrient supply (Best, 2019). Thus, including more detailed lake and reservoir process representations in the model would therefore improve the accuracy of nutrient load estimates and better capture the spatial and temporal variability in nutrient enrichment within river basins, especially rivers that are dammed.

*We intend to include this discussion in the revised manuscript.*

4. I suggest the authors to check the citations used.

   o In Section 2.3, several global water quality models are mentioned, including DynQual, MARINA, and IMAGE-GNM. However, I recommend the authors to check the citations. For IMAGE-GNM, the model version of Beusen et al., (2015) is cited, while a more updated version exist as described in Beusen et al., (2022). For the MARINA model, the calibrated model version of Strokal et al., (2016) is cited, while the authors refer to the uncalibrated version (e.g. Micella et al., (2024).

**Response**: Thank you for your suggestions. Actually, for the IMAGE-GNM model, with suggestions from the authors, we will revise IMAGE-GNM to IMAGE-DGNM and use both citations i.e. (Beusen et al., 2015; Beusen et al., 2022).

*So we will correct this and align the citations in the revised manuscript.*

   o In Section 3.1, the authors compare the global TN export estimates of CoSWAT-WQ with other global models. Here, I also suggest the co-authors to check the values and the citations. For example, according to the manuscript estimates for total N export include "39.1 Tg/yr from IMAGE-GNM", whereas Table 1 in Beusen et al. (2022) reports 41 Tg/yr. Perhaps the authors have excluded aquaculture for comparison purposes. If this is the case, I suggest the authors specify this. Next to the IMAGE-GNM model, estimates of the MARINA model are provided "34.8 Tg/yr from the MARINA model (Strokal et al., 2021)" whereas Strokal et al., (2021) report only inputs to rivers from urban sources.

**Response**: Thank you for spotting this. We will rectify the citations and cross-check the values.

5. Lastly, I would like to suggest some textual changes:

   o The title refers to a "community global SWAT+ water quality model". Yet, the word "community" does not appear in the rest of the manuscript. In the Conclusion, the authors refer to the free accessibility and highly customizable aspects of the model. I suggest the authors to integrate the word 'community model' somewhere to strengthen this message.

**Response**: Thank you for spotting this. We will incorporate more the word "community model" in the manuscript text to strengthen the message.

o    The abbreviations for N and P are introduced multiple times in the Introduction section; the same applies to HRU. I suggest only introducing them once.

**Response**: Thank you for the suggestion. We will rectify the introductions of N, P and HRUs.

o    Line 68, "This makes physically based model approaches…", could be rephrased to make the connection between process-based models and physically based models more clear.

**Response**: Thank you for the suggestion. We will rephrase the sentence to connect process-based and physically based models more clearly.

o    Line 82, "Despite being data-intensive, SWAT(+) models can now benefit from…", I would start the sentence with "SWAT(+) models can now benefit from….", as the first part may cause confusion due to the aforementioned lack of data and the computational intensity already becomes clear from the rest of the sentence.

**Response**: Thank you for the suggestion. We will rephrase the sentence as suggested.

o    Line 80, "application" to "applications"

**Response**: Thank you for the suggestion. We will rectify this in the revised manuscript.

o    Lines 90-91, "to identify hotspots and trends", does this refer to seasonal trends or future trends? Or both?

**Response**: Thank your questions. It refers to both. We will make this clear in the revised manuscript.

o    "CoSWAT-WQ" or "COSWAT-WQ" both spellings are used

**Response**: Thank you for spotting this. We will harmonize the wording in the revised manuscript.

o    "nonpoint" and "non-point" are both used, choose one to stay consistent

**Response**: Thank you for spotting this. We will harmonize the wording in the revised manuscript.

o    Section 2.1 is titled "SWAT+ model description", change to "CoSWAT-WQ model description"? As I would expect a description of the newly developed model.

**Response**: Thank you for the suggestion. We will rephrase the sentence as suggested.

o    Line 156, "was extracted" to "were extracted"

**Response**: Thank you for the suggestion. We will rephrase the sentence as suggested.

o    Lines 259-261, "Hydrology plays a particularly important role in model results, though this discussion focuses on nutrient-related processes. Particularly, there is a need to improve river flow modelling and to incorporate water management features such as lakes and reservoirs, as highlighted in Chawanda et al. (2025)." The use of the word 'particularly' feels a bit odd here, as the previous sentence highlights the focus on nutrient-related processes.

**Response**: Thank you for the suggestion. We will rephrase the sentence as suggested in the revised manuscript and remove the word "particularly".

References included in this review [all references suggested here are already included in the manuscript or are linked to water quality models that have already been referred to in the manuscript]

- Nkwasa, A., Chawanda, C. J., Jägermeyr, J., & Van Griensven, A. (2022). Improved representation of agricultural land use and crop management for large-scale hydrological impact simulation in Africa using SWAT+. Hydrology and Earth System Sciences, 26(1), 71-89.

- Jones, E. R., Graham, D. J., van Griensven, A., Flörke, M., & van Vliet, M. T. (2024). Blind spots in global water quality monitoring. Environmental Research Letters, 19(9), 091001.

- Neitsch, S.L., Arnold, J.G., Kiniry, J.R., Williams, J.R., King, K.W., 2005. SWAT theoretical documentation. Soil Water Res. Lab. Grassl. 494, 234–235.

- Beusen, A. H., & Bouwman, A. F. (2022). Future projections of river nutrient export to the global coastal ocean show persisting nitrogen and phosphorus distortion. Frontiers in Water, 4, 893585.

- Micella, I., Kroeze, C., Bak, M. P., Tang, T., Wada, Y., & Strokal, M. (2024). Future scenarios for river exports of multiple pollutants by sources and sub-basins worldwide: Rising pollution for the Indian Ocean. Earth's Future, 12(11), e2024EF004712.

**Citation**: https://doi.org/10.5194/egusphere-2025-703-RC1

**References:**

Arnold, J.G., Kiniry, J.R., Srinivasan, R., Williams, J.R., Haney, E.B., Neitsch, S.L., 2013. SWAT 2012 input/output documentation. Texas Water Resources Institute.

Arnold, J.G., Moriasi, D.N., Gassman, P.W., Abbaspour, K.C., White, M.J., Srinivasan, R., Santhi, C., Harmel, R.D., Van Griensven, A., Van Liew, M.W., 2012. SWAT: Model use, calibration, and validation. Trans. ASABE 55, 1491–1508. https://doi.org/doi:10.13031/2013.42256

Best, J., 2019. Anthropogenic stresses on the world's big rivers. Nat. Geosci. 12, 7–21. https://doi.org/10.1038/s41561-018-0262-x

Beusen, A.H.W., Doelman, J.C., Van Beek, L.P.H., Van Puijenbroek, P.J.T.M., Mogollón, J.M., Van Grinsven, H.J.M., Stehfest, E., Van Vuuren, D.P., Bouwman, A.F., 2022. Exploring river nitrogen and phosphorus loading and export to global coastal waters in the Shared Socio-economic pathways. Glob. Environ. Change 72, 102426. https://doi.org/10.1016/j.gloenvcha.2021.102426

Beusen, A.H.W., Van Beek, L.P.H., Bouwman, L., Mogollón, J.M., Middelburg, J.B.M., 2015. Coupling global models for hydrology and nutrient loading to simulate nitrogen and phosphorus retention in surface water–description of IMAGE–GNM and analysis of performance. Geosci. Model Dev. 8, 4045–4067.

Gan, J., Wang, X., Yuan, Q., Xing, X., Liu, S., Du, C., Zheng, Y., Liu, Y., 2025. Impact of damming on nutrient transport and transformation in river systems: A review. Water Sci. Eng. 18, 209–220. https://doi.org/10.1016/j.wse.2024.11.001

Hu, C., Sadras, V.O., Lu, G., Zhang, P., Han, Y., Liu, L., Xie, J., Yang, X., Zhang, S., 2021. A global meta-analysis of split nitrogen application for improved wheat yield and grain protein content. Soil Tillage Res. 213, 105111. https://doi.org/10.1016/j.still.2021.105111

Jones, E.R., Bierkens, M.F.P., Wanders, N., Sutanudjaja, E.H., van Beek, L.P.H., van Vliet, M.T.H., 2023. DynQual v1.0: a high-resolution global surface water quality model. Geosci. Model Dev. 16, 4481–4500. https://doi.org/10.5194/gmd-16-4481-2023

Jones, E.R., Graham, D.J., Griensven, A. van, Flörke, M., Vliet, M.T.H. van, 2024. Blind spots in global water quality monitoring. Environ. Res. Lett. 19, 091001. https://doi.org/10.1088/1748-9326/ad6919

Kumar, R., Heße, F., Rao, P.S.C., Musolff, A., Jawitz, J.W., Sarrazin, F., Samaniego, L., Fleckenstein, J.H., Rakovec, O., Thober, S., Attinger, S., 2020. Strong hydroclimatic controls on vulnerability to subsurface nitrate contamination across Europe. Nat. Commun. 11, 6302. https://doi.org/10.1038/s41467-020-19955-8

Leff, B., Ramankutty, N., Foley, J.A., 2004. Geographic distribution of major crops across the world. Glob. Biogeochem. Cycles 18. https://doi.org/10.1029/2003GB002108

Micella, I., Kroeze, C., Bak, M.P., Strokal, M., 2024. Causes of coastal waters pollution with nutrients, chemicals and plastics worldwide. Mar. Pollut. Bull. 198, 115902. https://doi.org/10.1016/j.marpolbul.2023.115902

Nkwasa, A., Chawanda, C.J., Jägermeyr, J., van Griensven, A., 2022. Improved representation of agricultural land use and crop management for large-scale hydrological impact simulation in Africa using SWAT+. Hydrol. Earth Syst. Sci. 26, 71–89. https://doi.org/10.5194/hess-26-71-2022

Nkwasa, A., James Chawanda, C., Theresa Nakkazi, M., Tang, T., Eisenreich, S.J., Warner, S., van Griensven, A., 2024. One third of African rivers fail to meet the 'good ambient water quality' nutrient targets. Ecol. Indic. 166, 112544. https://doi.org/10.1016/j.ecolind.2024.112544

Virro, H., Amatulli, G., Kmoch, A., Shen, L., Uuemaa, E., 2021. GRQA: Global River Water Quality Archive. Earth Syst. Sci. Data 13, 5483–5507. https://doi.org/10.5194/essd-13-5483-2021

Yin, Y., Yang, K., Gao, M., Wei, J., Zhong, X., Jiang, K., Gao, J., Cai, Y., 2024. Declined nutrients stability shaped by water residence times in lakes and reservoirs under climate change. Sci. Total Environ. 953, 176098. https://doi.org/10.1016/j.scitotenv.2024.176098

Zhao, X., Zhou, Y., Min, J., Wang, S., Shi, W., Xing, G., 2012. Nitrogen runoff dominates water nitrogen pollution from rice-wheat rotation in the Taihu Lake region of China. Agric. Ecosyst. Environ. 156, 1–11. https://doi.org/10.1016/j.agee.2012.04.024

---

## Author Comment (AC2)

**Note:** The comments from the reviewer are in black while the responses from the authors are in blue.

**Reviewer 2**

Overall, the paper represents an important advance in global water quality modeling using SWAT+, offering a solid foundation for future community-driven improvements. However, uncertainty, representation of in-stream processes, and data generalization remain key challenges. My comments mainly concern the quality and scope of the validation.

**Response:** We sincerely thank the reviewer for their positive feedback and welcoming our work. We appreciate the recognition of the potential and significance of the CoSWAT-WQ model for advancing global water quality research. We also acknowledge the concerns regarding the discussion of uncertainty and validation, and we will address these points carefully in the revised manuscript.

I feel that the authors primarily consider spatial validation, comparing modeled and observed nutrient loads across catchments. At the same time, temporal validation (e.g. monthly or seasonal nutrient concentrations) is largely absent. In my opinion, uncertainty analysis is also sorely lacking. While some indicators are reported, uncertainty in input data (especially for fertilizers/manures and point sources) is insufficiently considered.

**Response:** Thank you for this valuable comment. While our original manuscript focused primarily on spatial validation, we agree that temporal validation and a more thorough uncertainty analysis could make the manuscript stronger. Although we included a monthly evaluation of nutrient concentrations (Fig. 8 and Supplementary Material A), we acknowledge that this aspect was not sufficiently discussed. In the revised manuscript, we will expand the discussion of temporal performance to better demonstrate the model's ability to simulate seasonal dynamics. Additionally, as the model relies on several global input datasets, this inherently introduces uncertainties into our simulations. While these datasets are essential for achieving global coverage, they come with limitations that can affect model accuracy in multiple ways. For example, point source nutrient loads, which are derived from global gridded datasets, also carry substantial uncertainty, especially because such data distribute emissions over grid cells, whereas in reality, wastewater discharges occur at specific point locations. This spatial generalization may lead to mismatches between modelled and actual nutrient input locations, particularly in urban or industrial regions. Similarly, fertilizer/manure data at a 0.5-degree resolution may fail to capture the heterogeneity of fertilizer/manure application rates. This can lead to misrepresentation of nutrient source areas at the local level. Similarly, using gridded climate data (e.g., precipitation) at the same resolution can significantly influence hydrological processes such as runoff generation, which are key drivers of nutrient mobilization and transport. These issues are further compounded in regions with sparse observational data, where global datasets rely more heavily on interpolation or modelling assumptions. As a result, model outputs in such areas may be less reliable in absolute terms, although they may still be useful for identifying relative patterns, trends, or hotspots. Thus, to improve the robustness of future simulations, regionally validated input datasets should be integrated wherever available.

*We will revise the manuscript to better discuss the temporal (monthly) evaluation and also expand on the uncertainties of the input data (fertilizer/manure and point sources).*

Regarding fertilizer and manure inputs, these are based on national statistics, which introduce large uncertainties in local management practices.

**Response:** Thank you for this valuable comment. Yes indeed, most of the crop management datasets we use are prepared from national statistics which inherently introduce uncertainties in local management practices. For example, crop management data at a 0.5-degree resolution may fail to capture the heterogeneity of agricultural practices, including crop types, fertilizer application rates, and management intensity. This can lead to misrepresentation of nutrient source areas at the local level. Unfortunately, to our knowledge, there are currently no globally consistent datasets with higher spatial resolution available for these inputs. However, one strength of our workflow is its flexibility, if higher-resolution data becomes available in the future, our approach can incorporate and reflect that added heterogeneity.

*We intend to expand our discussion of these limitations and address the implications of the current global management datasets.*

In the paper, the authors do not assess the sensitivity of nutrient results to stream process parameters - a significant gap in water quality modelling at this scale.

**Response:** Thank you for your insightful comment. We agree that instream process parameters are an important but often underexplored aspect in global water quality modelling. In this version of the model, we do not explicitly assess the influence of instream parameters, which we recognize as a limitation. This is an area we intend to explore in future model versions, particularly as we begin calibrating different components of the system. Beyond this study, a likely reason for the limited attention to instream process sensitivity or calibration in large-scale nutrient models is the inverse relationship between nutrient retention potential and stream depth. This means that smaller streams, which offer greater solute-benthos contact, tend to have higher nutrient retention capacities than larger rivers (Ye et al., 2012; Ensign and Doyle, 2006). As a result, input sources may have a more dominant influence on nutrient loads at larger scales, possibly leading to a reduced emphasis on instream processes. While our study did not explore this aspect in detail, we agree it could be significant, particularly at local or regional scales.

*We will make sure to note this limitation more clearly in the revised manuscript.*

The authors assume that the SWAT+ structure and parameters are universally applicable - but in reality, hydrological and biogeochemical processes vary with climate, soil, and management. For example, the same set of parameters may perform poorly in tropical systems but well in temperate ones. This is not explained.

**Response**: Thank you for this important observation. We agree that a discussion on the use of default model parameters is necessary, especially given the variability in hydrological and biogeochemical processes across different climatic, soil, and land management contexts. To give a bit of context here, SWAT+ can still provide reasonable outputs in ungauged basins when default or parameters are used. For example; Chen et al. (2023) and Niraula et al. (2011) compared critical source areas (CSAs) for sediment and nutrients of a calibrated and uncalibrated SWAT model in southwest China and southeast U.S respectively. Both studies showed that CSAs locations had high similarity (81–93%) with and without calibration. These studies show that the SWAT model can be applied to identify pollution hotspots without calibration in watersheds especially data poor cases. That said, we fully acknowledge that the default parameter set is not universally optimal as performance can vary

substantially between tropical and temperate systems, for example. Calibration remains essential wherever observational data are available, as it significantly improves model performance and reliability.

*We will clarify this point in the revised manuscript and expand the discussion to reflect on the use of default parameters and their subsequent implications.*

**Citation**: https://doi.org/10.5194/egusphere-2025-703-RC2

**References**

Chen, M., Janssen, A.B.G., de Klein, J.J.M., Du, X., Lei, Q., Li, Y., Zhang, T., Pei, W., Kroeze, C., Liu, H., 2023. Comparing critical source areas for the sediment and nutrients of calibrated and uncalibrated models in a plateau watershed in southwest China. J. Environ. Manage. 326, 116712. https://doi.org/10.1016/j.jenvman.2022.116712

Ensign, S.H., Doyle, M.W., 2006. Nutrient spiraling in streams and river networks. J. Geophys. Res. Biogeosciences 111. https://doi.org/10.1029/2005JG000114

Niraula, R., Kalin, L., Wang, R., Srivastava, P., 2011. Determining nutrient and sediment critical source areas with SWAT: effect of lumped calibration. Trans. ASABE 55, 137–147.

Ye, S., Covino, T.P., Sivapalan, M., Basu, N.B., Li, H., Wang, S., 2012. Dissolved nutrient retention dynamics in river networks: A modeling investigation of transient flows and scale effects. Water Resour. Res. 48, 2011WR010508. https://doi.org/10.1029/2011WR010508

---

## Author Comment (AC3)

**Note:** The comments from the community are in black while the responses from the authors are in blue.

Dear authors,

I would like to respectfully raise a number of concerns regarding our ability – as a hydrological modelling community – to predict water quality at the global scale. Some are generic; some are more specific to this manuscript.

**Response**: We sincerely thank Dr. Tobias Krueger for taking the time to read our manuscript and share his insightful concerns. We appreciate his thoughtful feedback and the opportunity to engage with the important questions he raises about global-scale water quality modelling.

**Representing the relevant processes at the global scale and having data to parameterise them**

Land management is known to be very important for modulating pollution transfers. But how to even define those sets of rules (decision tables) for scheduling management at the 2x2km grid scale or HRU scale (page 5)? How do decisions made at the farm scale or smallholder scale aggregate? How to represent heterogeneity?

**Response:** Thank you for the question. To explain this simply, consider an agricultural grid or hydrologic response unit (HRU), as shown in Figure R1. Above each of these grids, we have several management layers, including crop type, cropping calendar, and fertilizer or manure application. For each agricultural grid or HRU, we read the corresponding values from these management layers and apply them accordingly. This process is pre-processed using a Python script, which organizes the information into a decision table (an example is shown in Figure R2), following the workflow developed in Nkwasa et al. (2022). This allows us to automate the assignment of agricultural management practices across all agricultural grids or HRUs.

[Figure]

Fertilizer/manure application

Cropping calendar (plant/harvest days)

Crop type

Grid/pixel/HRU level

**Fig. R1**: Schematic representation of an agricultural grid or hydrologic response unit (HRU) and its associated management layers

[Figure]

**Fig. R2**: Sample decision table showing the planting and harvesting of an agricultural crop

The question of farm or smallholder scale depends on the resolution of the datasets used for the management layers. In our study, we use freely globally available datasets, most of which are at a resolution of 0.5 degrees. Since our model operates at 2 km grid resolution, multiple 2 km grid cells fall within each 0.5-degree cell. This means that all agricultural grid cells within a single 0.5-degree grid share the same management practices.

In summary, the degree of heterogeneity we can capture depends on the resolution of the agricultural input datasets. To our knowledge, there are currently no global crop management datasets available at a higher resolution than those we have used. However, one strength of our workflow is its flexibility, if higher-resolution data become available in the future, this approach can incorporate and reflect that added heterogeneity.

*We intend to add this short workflow description in the SI material.*

What information exists to estimate tile drainage globally (figure 2)?

**Response**: We do not factor in tile drainage in our global study and we do not expect the model to perform so well in areas with a lot of artificially drained agricultural systems like in the Midwest United States and selected parts of Europe  The only existing global dataset on drained areas, developed by Feick et al. (2005), presents a map depicting the fraction of drained land within 5 x 5 minute raster cells. While this map is useful when modelling global water resources, its still not suitable for use in model applications requiring area-differentiated delineation of drained land (Tetzlaff et al., 2009). This map also only show estimates of the area of land that is drained, or the area of land that is likely to be densely drained, but do not provide information on drain design characteristics or specific density metrics which are key in implementing tile drains. Additionally, there are no appropriately scaled estimates of nutrient delivery through tile drainage to compare with model results.

*We intend to add a paragraph to the discussion which will highlight this limitation and bring it out as an area for both future model and data development under "section 4.2 Current missing links and future direction"*

How do the uncertainties in global data (which are often generated themselves from models), e.g. on point sources, plant and harvest dates, and fertilizer and manure use rates, affect model predictions (page 6)?

**Response**: Thank you for this question. We agree that uncertainties in model structure and global input data such as point sources, planting and harvest dates, and fertilizer/manure application rates can significantly influence model outputs. These data are often derived from a combination of remote sensing, national statistics, and modelling approaches, each carrying inherent uncertainties due to spatial heterogeneity, spatial and temporal

resolutions, reporting gaps, or even assumptions in upscaling methods. We acknowledge that these uncertainties can propagate through the model and affect both the magnitude and spatial patterns of predicted nutrient loadings and that such uncertainties are especially prevalent when modelling at large spatial extents.

*We intend to add a paragraph to the discussion which will acknowledge the uncertainties and reflect on the implications for interpreting large-scale water quality modelling results.*

**Choosing an appropriate tool for what we want to predict**

When it is said on page 13 that the primary objective of global water quality models is not to predict exact daily concentrations, then maybe a model that aims to represent processes at that scale is not appropriate. Monthly aggregated data are not predicted well by the daily model either, see below.

When the aim is instead to identify spatial and temporal pollution hotspots within global river networks (page 13), there may be other data-based methods that are better suited for this purpose.

**Response**: Thank you for this valuable comment. We agree that model selection should be driven by the intended purpose of the study. While this study does not aim to predict exact daily concentrations, but rather to initiate a global water quality model with the capacity to be applied at multiple temporal scales (from daily to annual), a process-based model like CoSWAT-WQ remains a suitable and effective choice given our objective. A key advantage of CoSWAT-WQ over more empirical or purely data-driven approaches is its ability to simulate cause-effect relationships between landscape processes and water quality outcomes. This allows us not only to identify where pollution occurs, but also to understand why it occurs by tracking the sources, pathways, and transformations of pollutants through the hydrological system. Furthermore, CoSWAT-WQ allows us to evaluate the effectiveness of different land and water management practices through scenario analysis, providing valuable insights for mitigating soil erosion and reducing nutrient loads from agricultural landscapes.

CoSWAT-WQ's semi-distributed structure captures spatial heterogeneity in land use, topography, and soil types, which is essential for modelling nutrient and sediment dynamics realistically across large and diverse catchments. Unlike black-box models, CoSWAT-WQ enables testing of land management scenarios by altering inputs or different cropping practices, making it a valuable tool for exploring mitigation/adaptation strategies and projecting future changes.

Additionally, we selected the CoSWAT-WQ model because SWAT is one of the most widely used water quality models across different spatial and temporal scales, supported by a large user community. This broad adoption means that the model is well-tested and familiar to many researchers, allowing others not only to contribute to its continued improvement but also to use parts of our global model such as individual subbasins for their own regional studies. We also anticipate that, with continued community involvement, simulations from this global water quality model will be further refined at higher temporal resolution (daily or even sub-daily).

*We intend to rephrase our objective to show that we to initiate a global water quality model with the capacity to be applied at multiple temporal scales (from daily to annual).*

**Model calibration**

The model in this study is said to be uncalibrated (page 6). How were parameter values determined instead?

**Response**: Thank you for your question. We used the default CoSWAT-WQ model parameters, which are based on literature and previous applications of the SWAT and SWAT+ models (Arnold et al., 2012; Arnold et al., 2013). As noted in the discussion (Line 270), future research will focus on model calibration and validation using observational data, especially from data-rich regions, to enhance the model's accuracy and reliability.

*We intend to state clearly that we use the default CoSWAT-WQ model parameters.*

The lack of sufficient observational data, strong spatial biases in available observations and uncertainties in input datasets (page 7) are put forward as obstacles to model calibration. What do these obstacles say about our ability to run a model at all?

**Response**: Thank you for your question. Indeed, these challenges are well-recognized and present significant obstacles to model calibration, particularly at the global scale. However, we believe these constraints should not preclude efforts to run global models. On the contrary, such modelling initiatives are essential for guiding future data collection, identifying priority areas for monitoring, and advancing methodological development. Our current approach focuses on identifying major spatial and temporal pollution hotspots, which can be cross-validated through model intercomparison efforts or supported by existing literature. While we acknowledge that uncertainties remain high, global models like this one, can still provide valuable insights especially for exploring scenarios and informing policy. Importantly, we see these modelling efforts as part of an iterative process. As more observational data and even remote sensing capabilities becomes available and modelling capacities continue to improve, we expect model performance and reliability to improve. We therefore view the current stage as an essential foundation for continuous development and improvement, rather than a limitation to modelling itself.

It is argued that the model is theoretical applicable in regions with no calibration data without a significant loss in performance (page 7). However, the hydrological modelling literature has repeatedly shown that even physically well-founded models require calibration to make them work in particular places. Among other things, this is because the processes that are represented are "effective" processes at the scale of application, either because physical theory is moved between scales or scale-bound abstractions are used in the first place (the MUSLE is an obvious example in this study).

**Response**: Thank you for this important observation. We fully agree that, as shown in a wide body of hydrological literature, including with SWAT, even physically based or semi-distributed models require calibration to effectively represent catchment-specific "effective" processes, especially when operating at larger spatial scales. Our statement on page 7 was not intended to suggest that calibration is unnecessary, but rather to reflect findings in the literature indicating that SWAT can still provide reasonable outputs in ungauged basins when default or parameters are used. For example; Chen et al. (2023) and Niraula et al. (2011) compared critical source areas (CSAs) for sediment and nutrients of a calibrated and uncalibrated SWAT model in southwest China and southeast U.S respectively. Both studies showed that CSAs locations had high similarity (81–93%) with and without calibration. These studies show that the SWAT model can be applied to identify pollution hotspots without calibration in watersheds especially data poor cases.

*We intend to revise our statement on page 7 to clarify that it does not suggest calibration is unnecessary, but rather reflects findings in the literature indicating that SWAT can still produce reasonable outputs in ungauged basins when default parameters are used.*

When it is said that uncalibrated physically-based models are often preferred for global change assessment on page 7, I am unclear whether this is because they are uncalibrated or because they are physically-based, and why exactly either of these features makes them preferrable. Keeping in mind the limits of physically-based models discussed above.

**Response**: Thank you for this question and sorry for the lack of clarity. Our intention was to highlight that physically-based models are preferred in global change assessments because they are physically-based, not because they are uncalibrated. Modelling approaches can broadly be categorized into statistical and physically-based models (Caissie, 2006). Statistical models estimate water quality variables using empirical relationships such as regression, stochastic methods, or machine learning based on available observations (Wanders et al., 2019). These methods can lead to satisfactory results; however, these statistical relationships need to be determined based on available observations. Physically-based models, on the other hand, use underlying physical relationships between water quality and drivers such as meteorological, hydrological, and socio-economic variables. Their advantage over statistical methods is most evident in ungauged basins and in assessing the impacts of global changes such as climate and socio-economic changes on water quality. Physically based models are particularly suited for these scenarios, as they rely on established physical processes to estimate nutrient levels from climatic, hydrological, and socio-economic inputs (Jones et al., 2023). This enables reliable predictions even under unobserved or future conditions, making them applicable in the context of global change. Of course, physically based models have limitations of large data requirements and larger computational requirements (Caissie, 2006).

*We intend to revise our statement on page 7 to clarify that it the intention is to highlight that physically-based models are preferred in global change assessments because they are physically-based and not because they are uncalibrated.*

**Model intercomparison**

If all models in an intercomparison are affected by the aforementioned shortcomings in process representation and data, can a model intercomparison really enhance confidence in large-scale water quality models as argued on page 7?

**Response**: Thank you for this question. While it's true that all models in an intercomparison may share some common limitations in process representation and data inputs, model intercomparison can still enhance confidence in large-scale water quality modelling. Model intercomparison is valuable not because it guarantees "truth" in estimates, but because it reveals robust patterns and structural uncertainties across models. When different models with different structural setups and assumptions converge on similar outputs, this consistency can enhance confidence in those specific results. Alternatively, when models diverge, the intercomparison highlights areas where uncertainty is greatest and where further research or data improvement is most needed. So while intercomparison cannot eliminate the fundamental limitations shared by models, it can still provide useful insights, highlighting consistencies across models, revealing the range of possible outcomes.

In this study, the log-scaling of the predictions in figure 5 make the discrepancies between the two models appear smaller than they are on the original scale, which are often an order of magnitude. When recalculating the $R^2$ statistic on the back-transformed data, $R^2$ decreases from 0.71 to about 0.41 for TP, for example. I am unclear why the same data appear twice with two different symbols in the graph.

I suggest that a model intercomparison would benefit from a deeper exploration of the mechanics of the individual models. What might explain the differences or similarities in predictions?

**Response**: Thank you for this question. We agree that the more detail is needed in the global model intercomparison between CoSWAT-WQ, IMAGE-DGNM (Beusen et al., 2015; Beusen et al., 2022) and MARINA-Multi (Micella et al., 2024). To expand more, model differences can be partially attributed to structural variations among the models, including differences in process representation and the estimation of diffuse and point source inputs under scenario assumptions. For instance; (i) MARINA-Multi operates at the sub-basin scale, IMAGE-DGNMIMAGE-GNM at a 0.5° grid resolution, and CoSWAT-WQ at the hydrologic response unit (HRU) level. (ii) CoSWAT-WQ does not simulate dynamic land-use change beyond 2010, considering only changes in fertilizer and manure application while holding land use constant. This assumption may lead to inconsistencies in nutrient estimates in regions experiencing substantial expansion of agricultural land or loss of natural land cover, which are captured by the other models. (iii) Both CoSWAT-WQ and MARINA-Multi do not account for nutrient legacies in groundwater, which are included in the IMAGE-DGN model. (iv) IMAGE-DGN model incorporates aquaculture as an anthropogenic nutrient, a factor not accounted for in the CoSWAT-WQ and MARINA-Multi models. Beyond the above highlighted differences, more distinctions in the main model assumptions and implementations of CoSWAT-WQ, IMAGE-DGN and MARINA-Multi are given in Table R1. Although these examples do not capture all model differences, such structural and process variations inevitably influence model outputs.

**Table R1:** An overview of the major assumptions and implementations of CoSWAT-WQ, IMAGE-DGNM (Beusen et al., 2015; Beusen et al., 2022) and MARINA-Multi (Micella et al., 2024) models

| Parameter | CoSWAT-WQ | IMAGE-DGNM | MARINA-Multi |
|---|---|---|---|
| Landuse | Fixed on landuse of 2010 (ISIMIP) | Dynamic landuse from IMAGE-DGNM | Fixed land use of 2010 |
| Point source | IMAGE-DGNM | IMAGE-DGNM | IMAGE-DGNM |
| Fertilizer | IMAGE-DGNM | IMAGE-DGNM | IMAGE-DGNM |
| Manure on crops | IMAGE-DGNM | IMAGE-DGNM | IMAGE-DGNM |
| Manure on grassland | No grassland | IMAGE-DGNM | IMAGE-DGNM |
| Atmospheric deposition (N) | ISIMIP | IMAGE-DGNM | IMAGE-DGNM |
| Uptake crops | CoSWAT-WQ | IMAGE-DGNM | IMAGE-DGNM |
| Crop types | Sugar cane, maize, soybean, wheat, banana | Upland crops, wetland rice and legumes | based on IMAGE-DGNM |
| Spatial calculation level | HRU | Grid cells of 0.5 * 0.5 degree | sub-basins |
| Hydrology | CoSWAT-WQ | IMAGE-DGNM - PCRGLOBWB | VIC (Variable Infiltration Capacity) |

| | | | model (van Vliet et al., 2016) |
|---|---|---|---|
| Legacy in groundwater (N) | No | Yes | No |
| Accumulation in soils (P) | yes | yes | no |
| N, P inputs from floodplains | no | yes | no |
| Aquaculture (N and P) | no | yes | no |

*We have summarized the differences between three global water quality models in a Table R1, and we will include this text and table discussing the model differences in the main text and supplementary material of the revised manuscript.*

**Model comparison with observations**

The model does not fit the monthly data well (as seen by the KGE values and figure 8). Data uncertainties are mobilised to explain mismatches but not instances of better fits (page 10). However, if the data are uncertain then the better fits might equally well be due to chance.

**Response**: Thank you for this comment. We agree that while mismatches between model output and observations are partially attributed to data uncertainty, we also recognize that apparent agreements may likewise result from uncertainty rather than actual model skill. This possibility is consistent with findings in the hydrological modelling literature (Kavetski et al., 2003; Beven, 2006), where both poor and seemingly good fits can be driven by uncertain or noisy data. This also applies to calibrated models where it is significantly possible for calibrated parameter values to compensate for different types of errors (structural or observational) (Beven, 2006).

*We intend to include this clarification in the discussion to provide a more balanced interpretation of model performance.*

The recognition on page 13 that a comparison with observations and other global nutrient models can identify which parts of the model can be improved but that it is difficult to determine specific areas for improvement to me points to a promising research programme.

**Response**: We appreciate the reviewer's recognition of uncertainties in the input data (e.g., land use, climate, hydrology, wastewater flows) and the inherent complexity of surface, subsurface, and instream process interactions, to inform a promising direction for future research. These factors collectively complicate the identification of specific areas for model improvement. Disentangling the relative contributions of these uncertainties and process complexities represents a critical initial step towards improving model structure and parameterization. This is actually be a promising and necessary direction for future research, which will ultimately enhance the robustness of global nutrient models.

The recognition on page 13 that hydrology plays a particularly important role in model results whereas the discussion focuses on nutrient-related processes points into the same direction. The hydrological model setup is in a parallel discussion at EGUsphere.

**Response:** Thank you for the comment. Indeed, the hydrological model setup is discussed in a parallel manuscript at EGUsphere, which has received a positive initial evaluation and is currently under revision. To clarify, the revision does not involve any structural changes or re-runs of the hydrological model; rather, it focuses on improving the contextual presentation and clarity of the model description.

I agree with the assessment on page 14 that model validation and calibration to improve accuracy and reliability are needed.

**Response**: We are in agreement and this is the focus of ongoing and future work in our model development.

I hope these comments are helpful for improving global water quality assessments.

**Response**: We appreciate the reviewer's thoughtful comments and hope that the revised manuscript will contribute meaningfully to improving global water quality assessments.

**References**

Arnold, J.G., Kiniry, J.R., Srinivasan, R., Williams, J.R., Haney, E.B., Neitsch, S.L., 2013. SWAT 2012 input/output documentation. Texas Water Resources Institute.

Arnold, J.G., Moriasi, D.N., Gassman, P.W., Abbaspour, K.C., White, M.J., Srinivasan, R., Santhi, C., Harmel, R.D., Van Griensven, A., Van Liew, M.W., 2012. SWAT: Model use, calibration, and validation. Trans. ASABE 55, 1491–1508. https://doi.org/doi:10.13031/2013.42256

Beusen, A.H.W., Doelman, J.C., Van Beek, L.P.H., Van Puijenbroek, P.J.T.M., Mogollón, J.M., Van Grinsven, H.J.M., Stehfest, E., Van Vuuren, D.P., Bouwman, A.F., 2022. Exploring river nitrogen and phosphorus loading and export to global coastal waters in the Shared Socio-economic pathways. Glob. Environ. Change 72, 102426. https://doi.org/10.1016/j.gloenvcha.2021.102426

Beusen, A.H.W., Van Beek, L.P.H., Bouwman, L., Mogollón, J.M., Middelburg, J.B.M., 2015. Coupling global models for hydrology and nutrient loading to simulate nitrogen and phosphorus retention in surface water–description of IMAGE–GNM and analysis of performance. Geosci. Model Dev. 8, 4045–4067.

Beven, K., 2006. A manifesto for the equifinality thesis. J. Hydrol., The model parameter estimation experiment 320, 18–36. https://doi.org/10.1016/j.jhydrol.2005.07.007

Caissie, D., 2006. The thermal regime of rivers: a review. Freshw. Biol. 51, 1389–1406. https://doi.org/10.1111/j.1365-2427.2006.01597.x

Chen, M., Janssen, A.B.G., de Klein, J.J.M., Du, X., Lei, Q., Li, Y., Zhang, T., Pei, W., Kroeze, C., Liu, H., 2023. Comparing critical source areas for the sediment and nutrients of calibrated and uncalibrated models in a plateau watershed in southwest China. J. Environ. Manage. 326, 116712. https://doi.org/10.1016/j.jenvman.2022.116712

Feick, S., Siebert, S., Döll, P., 2005. A digital global map of artificially drained agricultural areas.

Jones, E.R., Bierkens, M.F.P., Wanders, N., Sutanudjaja, E.H., van Beek, L.P.H., van Vliet, M.T.H., 2023. DynQual v1.0: a high-resolution global surface water quality model. Geosci. Model Dev. 16, 4481–4500. https://doi.org/10.5194/gmd-16-4481-2023

Kavetski, D., Franks, S.W., Kuczera, G., 2003. Confronting Input Uncertainty in Environmental Modelling, in: Calibration of Watershed Models. American Geophysical Union (AGU), pp. 49–68. https://doi.org/10.1029/WS006p0049

Micella, I., Kroeze, C., Bak, M.P., Strokal, M., 2024. Causes of coastal waters pollution with nutrients, chemicals and plastics worldwide. Mar. Pollut. Bull. 198, 115902. https://doi.org/10.1016/j.marpolbul.2023.115902

Niraula, R., Kalin, L., Wang, R., Srivastava, P., 2011. Determining nutrient and sediment critical source areas with SWAT: effect of lumped calibration. Trans. ASABE 55, 137–147.

Nkwasa, A., Chawanda, C.J., Jägermeyr, J., van Griensven, A., 2022. Improved representation of agricultural land use and crop management for large-scale hydrological impact simulation in Africa using SWAT+. Hydrol. Earth Syst. Sci. 26, 71–89. https://doi.org/10.5194/hess-26-71-2022

Tetzlaff, B., Kuhr, P., Wendland, F., 2009. A new method for creating maps of artificially drained areas in large river basins based on aerial photographs and geodata. Irrig. Drain. 58, 569–585. https://doi.org/10.1002/ird.426

van Vliet, M.T.H., van Beek, L.P.H., Eisner, S., Flörke, M., Wada, Y., Bierkens, M.F.P., 2016. Multi-model assessment of global hydropower and cooling water discharge potential under climate change. Glob. Environ. Change 40, 156–170. https://doi.org/10.1016/j.gloenvcha.2016.07.007

Wanders, N., van Vliet, M.T.H., Wada, Y., Bierkens, M.F.P., van Beek, L.P.H. (Rens), 2019. High-Resolution Global Water Temperature Modeling. Water Resour. Res. 55, 2760–2778. https://doi.org/10.1029/2018WR023250